# Active DNA damage response signaling initiates and maintains meiotic sex chromosome inactivation

Hironori Abe [1,2,4] ✉, Yu-Han Yeh [1,4], Yasuhisa Munakata [1], Kei-Ichiro Ishiguro [2], Paul R. Andreassen [3] & Satoshi H. Namekawa [1] ✉

Meiotic sex chromosome inactivation (MSCI) is an essential process in the male germline. While genetic experiments have established that the DNA damage response (DDR) pathway directs MSCI, due to limitations to the experimental systems available, mechanisms underlying MSCI remain largely unknown. Here we establish a system to study MSCI ex vivo, based on a short-term culture method, and demonstrate that active DDR signaling is required both to initiate and maintain MSCI via a dynamic and reversible process. DDR-directed MSCI follows two layers of modifications: active DDR-dependent reversible processes and irreversible histone post-translational modifications. Further, the DDR initiates MSCI independent of the downstream repressive histone mark H3K9 trimethylation (H3K9me3), thereby demonstrating that active DDR signaling is the primary mechanism of silencing in MSCI. By unveiling the dynamic nature of MSCI, and its governance by active DDR signals, our study highlights the sex chromosomes as an active signaling hub in meiosis.

Sex chromosomes are a classical model of chromosome-wide gene regulation. There are two distinct forms of chromosome-wide silencing in mammals; one is X-chromosome inactivation in females[1–4], and the other is meiotic sex chromosome inactivation (MSCI), an essential hallmark event in male germline development[1,5–8]. The mechanism underlying female X-inactivation has been extensively studied using an established ex vivo system[2,3]. In contrast, an ex vivo system that recapitulates the process of MSCI in culture remains to be established.

In pachytene spermatocytes during male meiosis, when autosomes complete chromosome synapsis, sex chromosomes, which do not synapse, are recognized and silenced to initiate MSCI[9–11]. This process is mediated by the DNA damage response (DDR) pathway, which otherwise recognizes DNA damage in somatic cells[12–14]. In meiosis, programmed double-strand breaks (DSBs) take place by the action of a DNA topoisomerase-like protein, SPO11, to initiate meiotic

recombination[15–18]. Thus, activation of the DDR, centered on the ATM serine/threonine-protein kinase, which recognizes DSBs, is inherent to normal meiosis and controls meiotic recombination and chromosome synapsis. Another serine/threonine-protein kinase, ATR, which is functionally related to ATM, also operates in a DDR pathway and regulates meiotic recombination[19–21]. ATR further directs the initiation of MSCI; at the onset of the pachytene stage, the DDR pathway centered on ATR mediates the phosphorylation of the histone variant H2AX at serine 139 (γH2AX) on the entire chromatin of the sex chromosomes (XY chromatin) and induces downstream histone post-translational modifications (PTMs)[14,22]. However, previous studies only used fixed mouse materials and yielded static pictures of MSCI; thus, the mechanism underlying MSCI remains largely elusive, in part, due to the lack of an ex vivo system that recapitulates the process of MSCI in culture.

[1]Department of Microbiology and Molecular Genetics, University of California, Davis, CA 95616, USA. [2]Department of Chromosome Biology, Institute of Molecular Embryology and Genetics (IMEG), Kumamoto University, Kumamoto 860-0811, Japan. [3]Division of Experimental Hematology & Cancer Biology, Cincinnati Children's Hospital Medical Center, University of Cincinnati College of Medicine, Cincinnati, OH 45229, USA. [4]These authors contributed equally: Hironori Abe, Yu-Han Yeh. ✉e-mail: habe@kumamoto-u.ac.jp; snamekawa@ucdavis.edu

During MSCI, ATR and γH2AX remain on XY chromatin during meiotic prophase I[13,23,24]. Counterintuitively, it has been reported, using a mouse model, that ATR is dispensable for the maintenance of MSCI; this is based on a genetic experiment in which normal MSCI was observed after the repression of the *Atr* transcript in meiosis[22]. In line with this observation, a repressive histone PTM, H3K9 trimethylation (H3K9me3), which is an evolutionarily conserved mark associated with meiotic silencing[25–27], was reported to be required instead for the initiation of MSCI[28]; this spawned the view that MSCI is a stable and irreversible process once gene silencing has been established[22,28]. Importantly, these potential mechanisms of MSCI have never been validated using ex vivo functional studies, and to date, there remains a mystery as to why DDR signaling remains on the XY chromosomes throughout meiotic prophase I.

To solve this conundrum, here we establish a system to study MSCI ex vivo. Using this ex vivo system, we show that active DDR signaling maintains MSCI, and that MSCI is dynamic and reversible. In other words, MSCI requires active and ongoing signaling by the DDR that is mediated by ATR. Further, we dissected downstream mechanisms and elucidated two layers of PTMs on meiotic sex chromosomes: active DDR-dependent reversible processes, including H2AX phosphorylation, and irreversible histone PTMs. Additionally, using a mouse model in which one of these irreversible marks, H3K9me3, is absent on the meiotic sex chromosomes, we demonstrate that the DDR initiates MSCI independent of H3K9me3. Thus, active DDR signaling initiates and maintains MSCI. Together with additional in vivo mouse analyses we performed, which confirm the dynamic nature of MSCI, our study reveals MSCI as a mode of chromosome regulation wherein active DDR signaling is the primary mechanism of silencing.

## Results

### Active DDR signaling is required for the maintenance of MSCI

While ATR is required for the initiation of MSCI, it was suggested that ATR is dispensable for the maintenance of MSCI[22]. We first sought to revisit the function of ATR in the maintenance of MSCI because active DDR signaling, detected with γH2AX, is retained on XY chromatin throughout meiotic prophase I after MSCI has been initiated[23]. To this end, we set out to develop a system to study MSCI ex vivo. We adapted a short-term culture method for pachytene spermatocytes[29,30], which was modified from a previous method[31]. Using this culture condition, the activity of ATR was controlled by treatment with an ATR inhibitor (ATRi) in testicular single-cell suspensions in culture (Fig. 1a). To specifically examine the maintenance of MSCI, we assessed only mid-late pachytene spermatocytes, in which MSCI has already been established. MSCI is initiated at the onset of the early pachytene stage, which lasts approximately 60 hours (60 h), and the subsequent mid-late pachytene stage lasts approximately 100 h[32]; the histone variant H1T, a nuclear marker, appears in mid-pachytene and persists into haploid spermatids[33]. This strategy excludes the analysis of spermatocytes that undergo initiation of MSCI during the 24 h time window of incubation with ATRi and thereby allows us to specifically examine the function of ATR on pachytene spermatocytes that have already established the γH2AX domain and which have initiated MSCI prior to treatment with ATRi (Supplementary Fig. 1). We first confirmed that spermatocytes treated with the ATRi AZ20[34] (5 or 10 μM) for 24 h displayed no loss of cell viability (Fig. 1b); thus, we can evaluate the effect of ATRi without any indirect effects caused by cell death. Importantly, under these same conditions, the γH2AX domain on the XY was clearly attenuated following treatment with 5 μM AZ20, and the γH2AX domain mostly disappeared with 10 μM AZ20 (Fig. 1c, d; Supplementary Fig. 2a), suggesting nearly complete suppression of ATR activity at the higher concentration. The specificity of ATR inhibition was confirmed by the normal presence of a pan nuclear γH2AX domain on the nuclei of leptotene spermatocytes, mediated by another PI3 kinase, ATM[35], after treatment with AZ20 (Supplementary Fig. 2b). We independently

validated the efficacy of ATRi using an alternative ATR inhibitor, AZD6738[36]; similar to results obtained with AZ20, the γH2AX domain on the XY disappeared, while cell viability was not changed after treatment with AZD6738 (Supplementary Fig. 2c, d). Because of the consistent effect of two independent ATR inhibitors, we focus on one of them, AZ20, in the experiments that follow. These results indicate that ATR activity is required for the maintenance of the γH2AX domain on the XY during the mid-late pachytene stage.

Next, we examined whether the maintenance of MSCI was altered by treatment with ATRi. Control spermatocytes showed signs of normal MSCI in which RNA polymerase II (Pol II) was excluded from the XY chromatin (Fig. 1e, f). By contrast, Pol II was present on the XY and was not excluded from the XY chromatin in spermatocytes after 24 h of treatment with AZ20 (Fig. 1e, f). This feature is concomitant with the disappearance of the γH2AX domain on the XY. To confirm the reversal of MSCI, we then performed gene-specific RNA Fluorescence in situ hybridization (FISH) to detect nascent transcripts of two X-linked genes (*Utx/Kdm6a* and *Lamp2* at 18 Mb and 37 Mb on the X chromosome, respectively (Fig. 1g, h) using our previously validated RNA FISH probes[37,38]. After 24 h of treatment with AZ20, *Utx* and *Lamp2* were actively transcribed in 87% and 98% of pachytene spermatocytes, while these genes were inactive in controls (Fig. 1g, h). Further, we performed Cot-1 RNA FISH, which was previously used to visualize global nascent transcription that is ceased in MSCI[13,39]. Cot-1 signals were excluded from XY chromatin in normal MSCI in controls, while they were not excluded from XY chromatin after treatment with ATRi (Supplementary Fig. 3). These results indicate that the maintenance of MSCI was altered by treatment with ATRi.

To further evaluate the transcriptional status of the sex chromosomes after treatment with ATRi, we next performed RNA-sequencing (RNA-seq). Because it is challenging to isolate pachytene spermatocytes after our small-scale culture using testicular single-cell suspensions, we instead isolated pachytene spermatocytes using Fluorescence-activated Cell Sorting (FACS) and cultured them using the same conditions as the experiments above (Fig. 2a and Supplementary Fig. 4a). Utilizing this approach, we confirmed the highly efficient removal of the γH2AX domain from the XY chromatin after treatment with ATRi (Supplementary Fig. 4b, c). We then performed RNA-seq in pachytene spermatocytes after 24 h culture with or without 10 μM AZ20, and found that the X and Y-linked genes were highly upregulated after treatment with ATRi (with 82.7- and 64.7- fold increases in mean expression from the X and Y chromosomes, respectively) compared to changes on the autosomes (Fig. 2b). Among 339 genes expressed on the X chromosome [≥1 TPM (transcript per million) in either experimental group], 247 genes were significantly upregulated (≥2-fold change and $P_{adj} < 0.05$) including *Utx* and *Lamp2*, while differentially expressed genes were barely detected on autosomes (Fig. 2c). Likewise, among 9 Y-linked genes, 5 genes were upregulated after treatment with ATRi (Fig. 2c). Together, these results demonstrate that active ATR signaling is required for the maintenance of MSCI, highlighting a strong association between γH2AX domain formation and MSCI. These results further validate the capacity of our ex vivo system to examine MSCI.

To dissect the ATR-dependent mechanism underlying the maintenance of MSCI, we next sought to determine the localization of ATR-related DDR factors that are required for the initiation of MSCI. Mechanistically, the initiation of MSCI is directed by the DDR pathway in two genetically separable steps (Fig. 3a). In the initial step of this mechanism, unsynapsed chromosome axes (XY axes in MSCI) are recognized by BRCA1, the protein product of breast cancer susceptibility gene 1[13,40,41], by the ATR activator TOPBP1[22,24,42], and by ATR. ATR phosphorylates H2AX at serine 139, which attracts MDC1, a γH2AX-binding partner[14]. The subsequent spread of γH2AX into a chromosome-wide domain (XY chromatin in MSCI) in the second step of MSCI is directed by MDC1 in a feedforward mechanism whereby

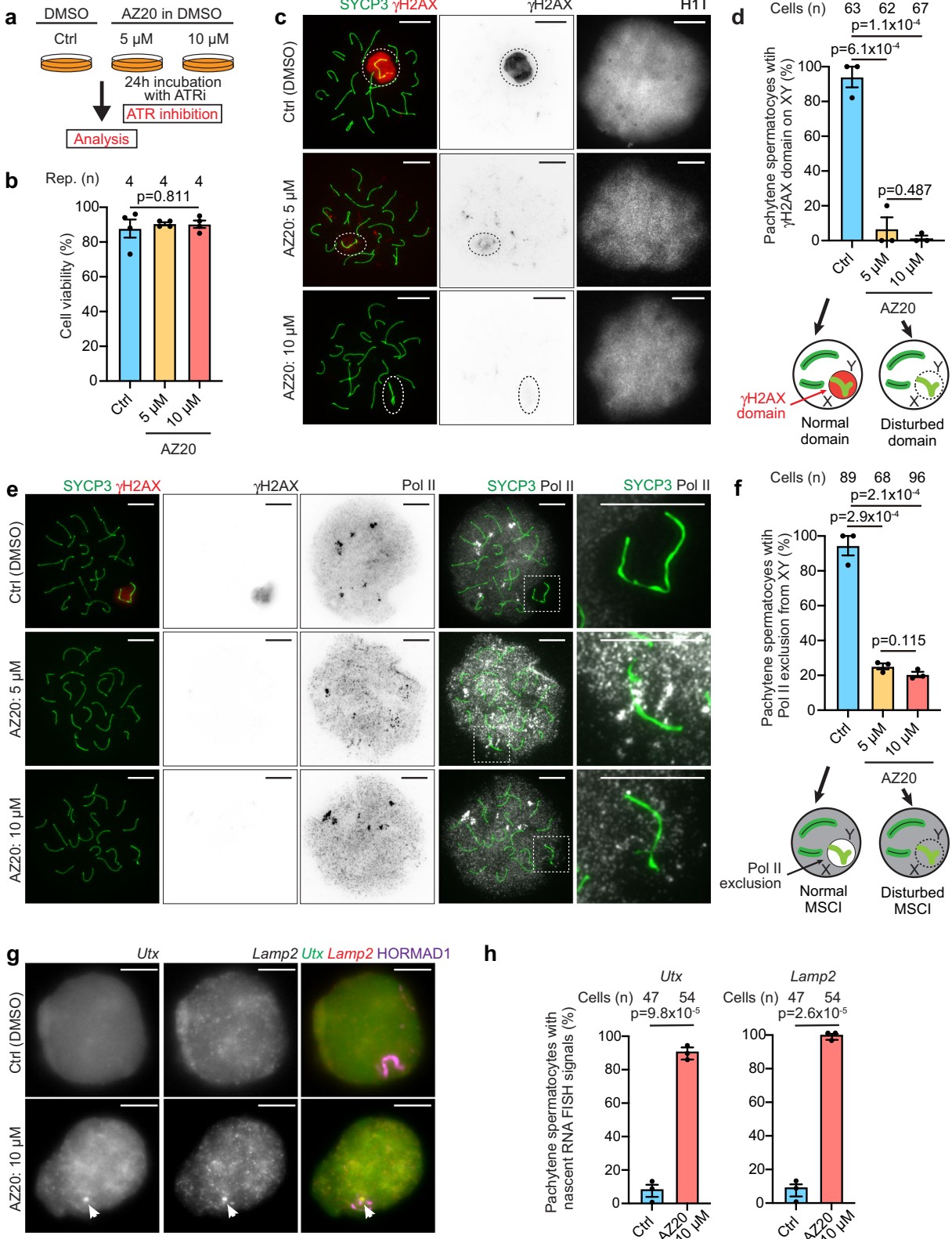

ATR and TOPBP1 are recruited to proximal nucleosomes, H2AX is again phosphorylated, MDC1 binds γH2AX, more ATR and TOPBP1 are incorporated, and so on[14] (Fig. 3a).

After 24 h of treatment with AZ20, BRCA1 remained localized only on unsynapsed axes and was unchanged as compared to the controls (Fig. 3b, f: additional images for Fig. 3 are shown in Supplementary Fig. 5). While TOPBP1 and ATR are intensely localized on unsynapsed

axes and spread to the XY chromatin in normal meiosis, TOPBP1 signals remained only on unsynapsed axes, and ATR signals mostly disappeared from both the unsynapsed axes and chromatin loops of the XY chromosomes, after AZ20 treatment (Fig. 3c, d, f). Additionally, in accord with the localization of γH2AX, signals for its binding partner, MDC1, largely disappeared from the entire XY chromatin after AZ20 treatment (Fig. 3e, f). These results indicate that

**Fig. 1 | Active ATR-dependent DDR signaling is required for the maintenance of MSCI. a** Schematic of the experimental design (ATRi: ATR inhibitor). **b** Viability of cultured cells after 24 h incubation shown as the mean ± s.e.m. for 4 independent experiments. **c** Chromosome spreads of mid-late pachytene spermatocytes immunostained with antibodies raised against SYCP3, which is a marker of meiotic chromosome axes, γH2AX, and H1T. XY chromosomes are indicated with dashed circles. **d** Quantification of mid-late pachytene spermatocytes with a normal γH2AX domain on the XY shown as the mean ± s.e.m. for 3 independent experiments. **e** Chromosome spreads of mid-late pachytene spermatocytes immunostained with antibodies raised against SYCP3, Pol II, and γH2AX. XY chromosomes are indicated with dashed squares and are magnified in the panels to the right. **f** Quantification of

mid-late pachytene spermatocytes with normal MSCI defined by Pol II exclusion shown as the mean ± s.e.m. for 3 independent experiments. **g** Gene-specific RNA FISH for X-linked *Utx* and *Lamp2* genes. XY axes were detected by immunostaining with antibodies raised against HORMAD1. RNA FISH signals (nascent transcripts) are indicated with arrowheads. **h** Quantification of mid-late pachytene spermatocytes with nascent RNA FISH signals for *Utx* (left panel) and *Lamp2* genes (right panel) shown as the mean ± s.e.m. for 3 independent experiments. Total numbers of analyzed nuclei are indicated in the panels (**d, f, h**). One-way ANOVA (**b**), two-tailed unpaired t-test (**d, f, h**). XY: XY chromosomes. Scale bars: 10 μm. Source data are provided as a Source Data file.

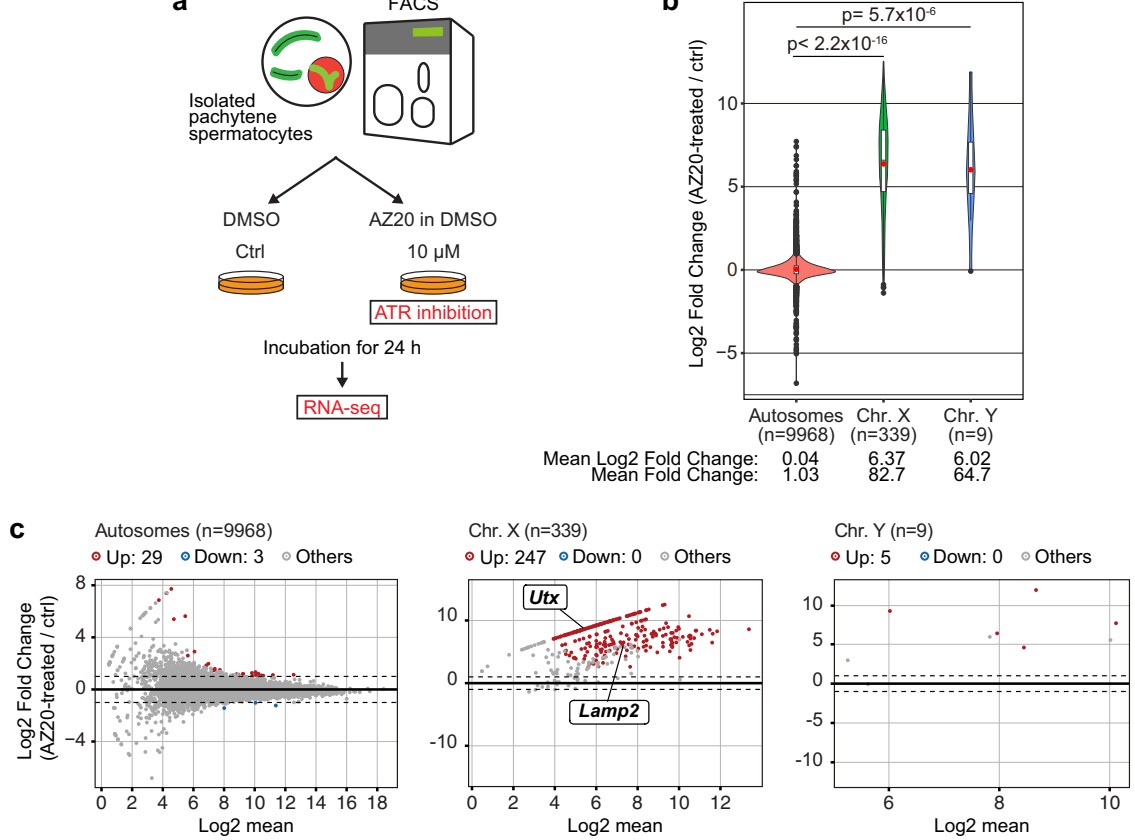

**Fig. 2 | Gene suppression under MSCI is directed by ATR-dependent DDR signaling. a** Schematic of the experimental design for RNA-seq analyses. **b** Violin plot with a Box plot overlay of the gene expression log2 fold change between AZ20-treated and ctrl spermatocytes. The central lines represent medians. The upper and lower hinges correspond to the 25th and 75th percentiles. The upper and lower whisker are extended from the hinge to the largest value no further than the 1.5x inter-quartile range (IQR) from the hinge. Log2 fold changes exceeding 1.5x IQR are shown by black dots. Red dots in the Box plot are the average of the log2 fold change. The two-sided *p*-values were calculated using a Wilcoxon rank sum test.

The total gene numbers analyzed are shown in the panel. **c** Scatter plots showing differentially expressed genes between AZ20-treated and ctrl spermatocytes. Differentially expressed genes, indicated as "Up" and "Down", were defined as those with a ≥ 2-fold change and $P_{adj} < 0.05$ using DESeq2. The log2 mean (*x*-axis) was calculated using the value of the baseMean from the DESeq2 analysis. The *Utx* and *Lamp2* genes shown in Fig. 1g, h were detected as derepressed genes by AZ20 treatment. Total numbers of genes analyzed are shown in each panel. Source data are provided as a Source Data file.

active ATR signaling is required for the maintenance of TOPBP1-ATR-MDC1 on chromatin loops to maintain MSCI. Taken together, these results demonstrate that active DDR signaling centered on ATR is not only required for the initiation of MSCI but is also essential for maintaining it.

**MSCI is reversible and its reestablishment depends on the DDR**

In addition to the requirement of DDR signaling for the initiation of MSCI, as indicated by previous genetic experiments using mouse models[12,14,22], results obtained by our culture model establish that active DDR signaling is also critical for the maintenance of MSCI. These results raised the possibility that MSCI can be manipulated based on

the activity of the DDR signaling pathway. To test this hypothesis, we examined whether MSCI can be reestablished via reactivation of ATR signaling following ATR inhibition. For this purpose, we first treated pachytene spermatocytes for 24 h with AZ20 at 5 or 10 μM, conditions which disrupt MSCI, and we then removed AZ20 from the culture medium to reactivate ATR signaling (Fig. 4a–c). After 3 h of incubation without AZ20, we found that a large γH2AX domain on XY chromatin reformed in the majority of H1T-positive pachytene spermatocytes similar to levels in spermatocytes never treated with AZ20 (Ctrl) (Fig. 4c, d). γH2AX intensity on XY chromosomes was decreased to a basal level after treatment with ATRi and recovered 3 h after removal of ATRi (Supplementary Fig. 6). Importantly, the reformed large

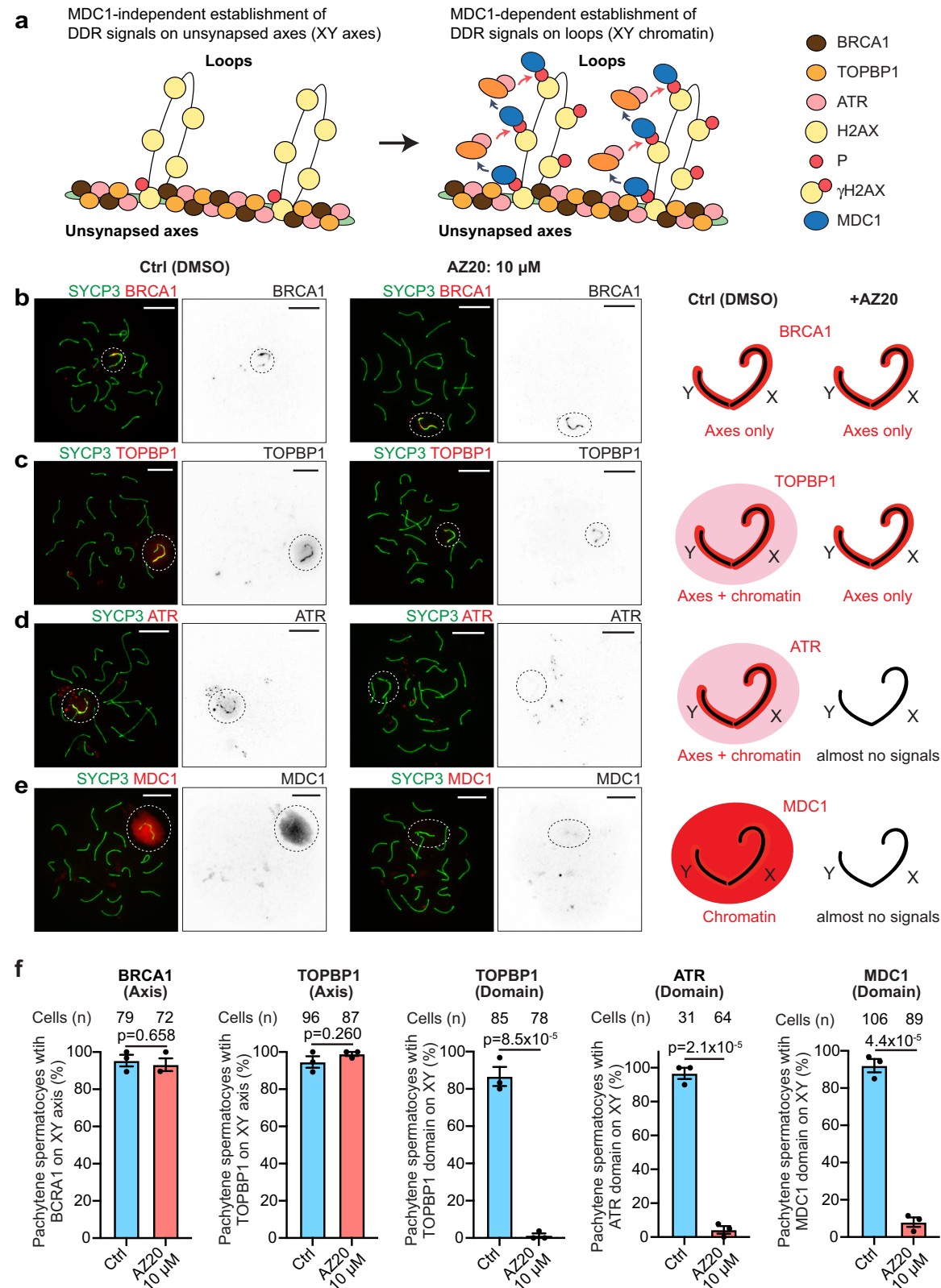

γH2AX domain on the XY chromatin excluded Pol II (Fig. 4e, f). This result unveiled an unexpected aspect of MSCI: MSCI is reversible and can be reestablished based upon recovery of ATR-dependent DDR signaling. Also, these results strengthen our findings in Fig. 1 that MSCI is maintained by ongoing ATR-dependent DDR signaling.

The recovery experiment from inhibition of ATR (Fig. 4e) suggests that the establishment of chromosome-wide silencing during MSCI is a

rapid process occurring within 3 h. It has been very difficult to capture spermatocytes with a partial γH2AX domain on XY chromatin during spermatogenesis in vivo[43,44], presumably because MSCI is a rapid DDR-dependent process. Therefore, it has been a challenge to capture the initial process of MSCI from classical analyses of mouse models. To circumvent this limitation, we took advantage of our culture method. To capture the reestablishment of MSCI, following 24 h of ATR

**Fig. 3 | The localization of critical DDR factors on the XY chromosomes, including TOPBP1, ATR and MDC1, is disrupted by inhibition of ATR.**
**a** Schematic of the machinery for γH2AX domain formation in pachytene spermatocytes. The unsynapsed XY axis is recognized by BRCA1, TOPBP1, and ATR. Further, γH2AX signals are established by ATR along the XY axis. γH2AX formation on the entire XY chromatin is mediated by a MDC1-dependent feedforward mechanism. **b**–**e** Chromosome spreads of mid-late pachytene spermatocytes immunostained with antibodies raised against SYCP3, BRCA1 (**b**), TOPBP1 (**c**), ATR

(**d**), and MDC1 (**e**). XY chromosomes are indicated with dashed circles. Localization patterns of DDR factors on the sex chromosomes with or without exposure to ATRi are depicted in the panels to the right. See Supplementary Fig. 5 for 5 μM data. **f** Quantification of mid-late pachytene spermatocytes with the indicated DDR factors on the XY chromosomes shown as the mean ± s.e.m. for 3 independent experiments. Total numbers of analyzed nuclei are indicated in the panels. Two-tailed unpaired t-test. Scale bars: 10 μm. Source data are provided as a Source Data file.

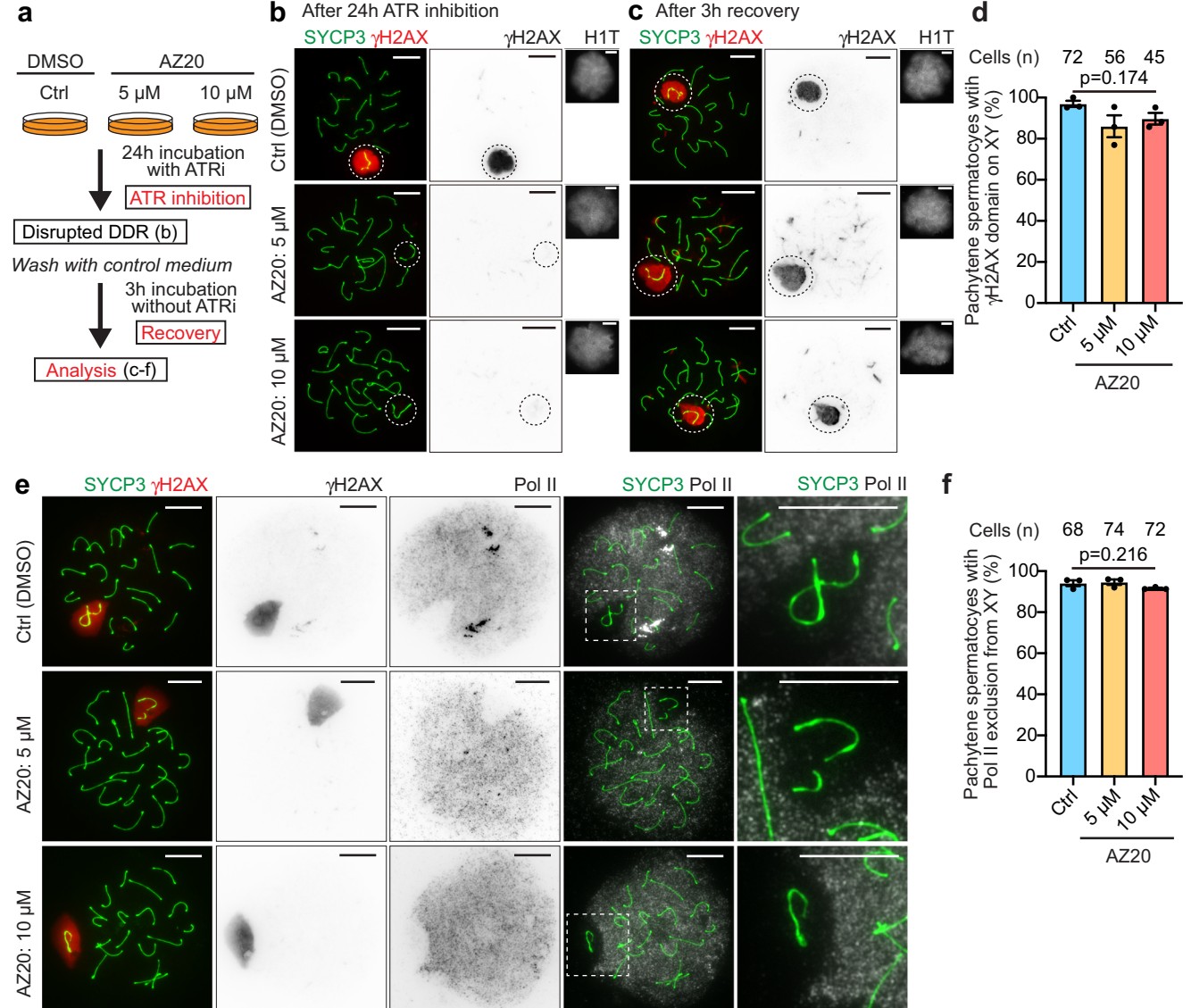

**Fig. 4 | DDR-dependent MSCI is reversible. a** Schematic of the experimental design. Spermatocytes were incubated for 3 h without ATRi, which was removed after a 24 h incubation with the ATRi AZ20. **b, c** Chromosome spreads of mid-late pachytene spermatocytes immunostained with antibodies raised against SYCP3, γH2AX, and H1T; after treatment with 5 or 10 mM AZ20 for 24 h (**b**), and after an additional 3 h without ATRi (**c**). XY chromosomes are indicated with dashed circles. Representative images of 3 independent experiments are shown. **d** Quantification of mid-late pachytene spermatocytes with normal γH2AX domains on the XY after recovery from ATRi, as determined in comparison to untreated spermatocytes,

shown as the mean ± s.e.m. for 3 independent experiments. **e** Chromosome spreads of mid-late pachytene spermatocytes immunostained with antibodies raised against SYCP3, Pol II, and γH2AX, demonstrating that Pol II is excluded from XY chromosomes after release from treatment with the ATRi AZ20. XY chromosomes are indicated with dashed squares and are magnified in the panels to the right. **f** Quantification of mid-late pachytene spermatocytes with normal MSCI defined by Pol II exclusion shown as the mean ± s.e.m. for 3 independent experiments. Total numbers of analyzed nuclei are indicated in the panels (**c, e**). One-way ANOVA (**c, e**). Scale bars: 10 μm. Source data are provided as a Source Data file.

inhibition, we tested a shorter time window of recovery of 30 min (Fig. 5a), in contrast to the 3 h recovery that showed complete reformation of the γH2AX domain on the XY (Fig. 4b, c). Remarkably, we found that partial establishment of the γH2AX domain took place on

XY chromatin at this intermediate time point of recovery (30 min) from ATR inhibition by 10 μM AZ20 (Fig. 5b). Notably, these partial γH2AX signals radiated out from unsynapsed axes, where DDR proteins such as BRCA1 and TOPBP1 were retained after ATR inhibition

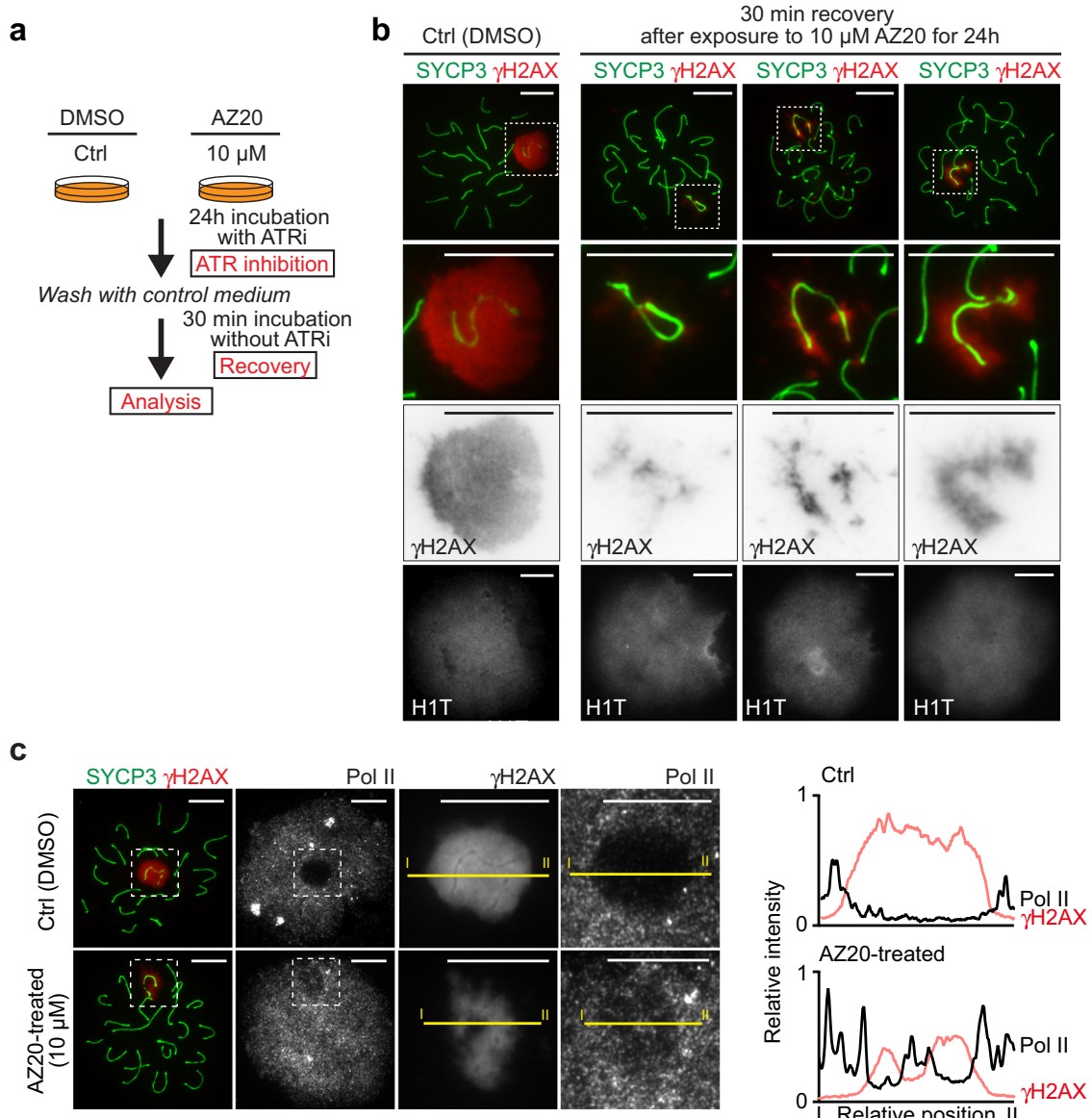

**Fig. 5 | ATR-dependent reestablishment of MSCI from unsynapsed axes.**
**a** Schematic of the experimental design. Following 24 h incubation with the ATRi AZ20, spermatocytes were additionally incubated for 30 min without ATRi.
**b, c** Chromosome spreads of mid-late pachytene spermatocytes harvested 30 min after release from a 24 h incubation with ATRi and immunostained with antibodies raised against SYCP3, γH2AX, and H1T (**b**) and SYCP3, Pol II, and γH2AX (**c**). Representative images of 3 independent experiments are shown. XY chromosomes are indicated with dashed squares and are magnified at the bottom (**b**) and the middle right and right (**c**). **c** Intensities of γH2AX and Pol II on yellow lines, marked with "I–II", were measured and are shown in the right-hand panels; the upper panel shows control and the bottom panel shows a spermatocyte harvested 30 min after release from a 24 h incubation with ATRi. Red: γH2AX, Black: RNA Pol II. Source data are provided as a Source Data file.

(Fig. 3). These results are in line with what was previously known about mechanisms of MSCI initiation: BRCA1 and TOPBP1 act upstream of ATR, and DDR signals subsequently spread from unsynapsed axes to XY chromatin at the onset of MSCI[14] (Fig. 3a). Importantly, we next examined whether these partial γH2AX domains are associated with silencing and found that these partial γH2AX signals locally excluded Pol II signals (Fig. 5c). Therefore, active DDR signaling is tightly coupled with Pol II exclusion, even in a narrow time window during γH2AX domain formation on XY chromatin. Together with the requirement for the DDR in the maintenance of MSCI, these results highlight a direct mechanistic link between active DDR signaling and silencing.

**The DDR directs both dynamic and irreversible events on the XY**
Following the initiation of MSCI, various PTMs are established on the sex chromosomes[8]. Given our discovery that maintenance of MSCI is

dependent on the DDR, we next sought to examine whether active DDR signaling is required for the maintenance of PTMs on the sex chromosomes. Again, during the initiation of MSCI, the DDR mediates two genetically separable steps: The first is the establishment of DDR signals along the XY axes, and the second is the MDC1-dependent amplification of γH2AX throughout the XY chromatin[14] (Figs. 3a, 6a). SUMO (small ubiquitin-related modifier) is among the earliest XY modifications at the onset of MSCI[45,46], and SUMOylation in MSCI is MDC1-dependent[14]. After AZ20 treatment for 24 h at 10 μM, SUMO1, a major member of the SUMO family, disappeared from the XY chromatin (Fig. 6b, e, additional pictures for Fig. 6 are shown in Supplementary Fig. 7). Although SUMOylation substrates on the XY chromatin have not been determined, these results suggest that SUMOylation is tightly coupled with active DDR signaling. Another PTM on XY chromatin, ubiquitination, is mediated by RNF8, a

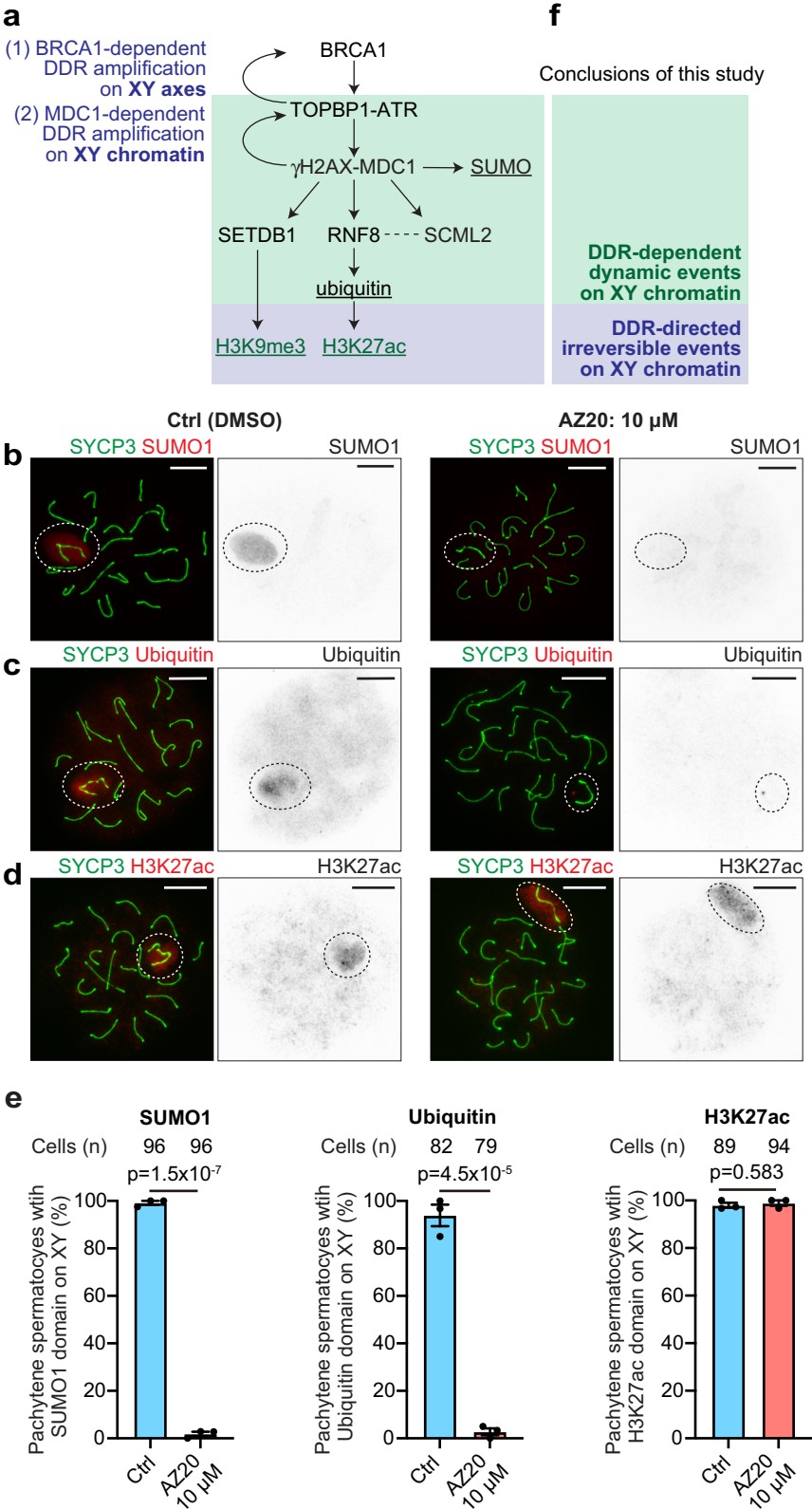

**Fig. 6 | Both dynamic and irreversible epigenetic states are established on the XY chromatin in a manner dependent on ATR-mediated signaling during pachytene. a** Schematic of the molecular pathways that establish various epigenetic states on the sex chromosomes. Post-translational modifications are underlined. **b–d** Chromosome spreads of mid-late pachytene spermatocytes immunostained with antibodies raised against SYCP3, SUMO1, ubiquitin, and H3K27ac. XY chromosomes are indicated with dashed circles. See Supplementary Fig. 7 for 5 µM data. Scale bars: 10 µm. **e** Quantification of mid-late pachytene spermatocytes with indicated histone modifications on the XY shown as the mean ± s.e.m. for 3 independent experiments. Two-tailed unpaired t-test. Total numbers of analyzed nuclei are indicated in the panels. **f** Schematic of the conclusions of this study. DDR-dependent dynamic events are shown in the green box, and DDR-directed irreversible events are shown in the blue box. Source data are provided as a Source Data file.

DDR-related E3 ubiquitin ligase that interacts with MDC1; downstream of RNF8-mediated ubiquitination, several histone PTMs, such as the active mark H3K27 acetylation (H3K27ac), are established[47–49] (Fig. 6a). This post-translational information supports "escape gene activation" by which a select set of sperm-related genes escape transcriptional repression in postmeiotic spermatids[47,48,50–52]. After AZ20 treatment, ubiquitin disappeared from the XY chromatin (Fig. 6c, e); however, H3K27ac remained on XY chromatin when ATR was inhibited (Fig. 6d, e). Thus, active DDR signaling is required for the maintenance of ubiquitination, but downstream of this, active DDR signaling is not required for the maintenance of H3K27ac once established.

To examine whether other histone PTMs are maintained independent of active DDR signaling, we next examined a repressive mark, H3K9me3, which is enriched on XY chromatin at the onset of MSCI in the early pachytene stage[28,32] and was reported to be required for the initiation of MSCI[28]. In normal meiosis, the distribution of H3K9me3 is dynamic; in the mid-pachytene stage, H3K9me3 largely disappears from X-chromatin, and is confined to X pericentric chromatin (X-PCH) and a part of the Y chromosome (Supplementary Fig. 8a). This pattern did not change in mid-pachytene spermatocytes after treatment with ATRi (Supplementary Fig. 8a). Therefore, active DDR signaling is not required to maintain the distribution of H3K9me3 on the sex chromosomes. Additionally, we tested the localization of SCML2, a germline-specific Polycomb protein and a component of XY chromatin that regulates epigenetic states and gene expression on the sex chromosomes[53,54]; we find that SCML2 localization is DDR-dependent, as revealed by incubating spermatocytes with ATRi (Supplementary Fig. 8b). Therefore, modifications on XY chromatin are largely dynamic, except for certain histone PTMs.

Together, our results unveil two distinct classes of PTMs on XY chromatin. Although the ATR-dependent DDR is required to establish all PTMs on XY chromatin[8], SUMOylation and ubiquitination are dynamic processes that require active DDR signaling to be maintained and thereby constitute one distinct class of PTMs on XY chromosomes during MSCI. In contrast, histone PTMs such as H3K27ac and H3K9me3 are irreversible events that do not require active DDR signaling to be maintained once they have been established and therefore represent another class of PTMs on the XY (Fig. 6f).

## DDR-mediated initiation of MSCI in the absence of H3K9me3

The above findings reveal that the DDR is tightly coupled with silencing but not with the status of H3K9me3. This raises a key question as to whether silencing in MSCI is directly regulated by the DDR or is instead mediated by a histone-based mechanism acting downstream of the DDR. To provide an answer to this question, we tested whether MSCI takes place in the absence of H3K9me3 on the XY. Since H3K9me3 is established on XY chromatin by SETDB1, to specifically deplete SETDB1 during male meiosis, we generated Setdb1 conditional knockout (Setdb1-cKO) mice[55] using Ddx4-Cre[56], which starts to express in germ cells after embryonic day 15 (E15) (Fig. 7a). Consistent with previous studies[28,57], we observed testicular defects, increased apoptosis, and meiotic arrest in Setdb1-cKO testes (Supplementary Figs. 9 and 10a). Additionally, we confirmed that, in Setdb1-cKO spermatocytes, SETDB1 was not present on the sex chromosomes (Supplementary Fig. 10b) and that H3K9me3 was absent on the sex chromosomes in most mutant cells (Fig. 7b, c). In littermate controls (Setdb1-ctrl), we observed a normal distribution of H3K9me3: H3K9me3 was present on XY chromatin at the early pachytene stage, and on X-PCH and a part of the Y at the mid-pachytene stage (Fig. 7c). Therefore, the dynamic pattern of H3K9me3 distribution from early- to mid- pachytene in spermatocytes is regulated in a SETDB1-dependent manner. The data shown in Fig. 7b also verify the highly efficient Cre-mediated deletion of the Setdb1-floxed allele by Ddx4-Cre.

In previous studies, the γH2AX domain was found to be present on the XY chromatin in Setdb1-cKO spermatocytes[28,57]. However, it

has not been determined whether the γH2AX domain on the XY chromatin in Setdb1-cKO spermatocytes is associated with silencing (i.e., whether the DDR is sufficient for silencing without H3K9me3). To test this, we performed Pol II immunostaining together with γH2AX and SYCP3 staining. In early-to-mid pachytene spermatocytes, the majority of mutant cells had an established γH2AX domain on XY chromatin (80% in the early pachytene stage and 87% in the mid pachytene stage: Fig. 7d, e; Supplementary Fig 11a). A minor population of mutant cells displayed disruption of the γH2AX domain, presumably due to chromosome synapsis errors (right bottom panel of Fig. 7d)[57]. Of note, γH2AX domain formation and exclusion of Pol II from XY chromatin were correlated in all cells examined; 100 % of early-to-mid pachytene spermatocytes with a γH2AX domain on the XY showed exclusion of Pol II from the γH2AX domain (Fig. 7d, e; Supplementary Fig. 11a). Further, as an indication of silencing associated with MSCI, we performed Cot-1 RNA FISH and confirmed that Cot-1 signals were excluded from the XY chromosomes in Setdb1-cKO spermatocytes that had normal synapsis of autosomes, as suggested by the presence of a single linear domain of HORMAD1 signal (Supplementary Fig. 11b, c); during meiosis, normally, autosomes become fully synapsed while the sex chromosomes remain unsynapsed, as indicated by the presence of a single linear HORMAD1 domain[58]. Likewise, nascent transcripts of Utx and Lamp2 were not detected in most of these Setdb1-cKO spermatocytes (Supplementary Fig. 11d, e). Together, these results indicate that the DDR is sufficient both for the initiation and maintenance of MSCI, including gene silencing, in the absence of H3K9me3; further, this supports the conclusion that active DDR signaling is the primary mechanism in the initiation and maintenance of MSCI.

## The DDR is required for the maintenance of MSCI in vivo

Finally, we sought to examine whether the DDR is required for the maintenance of MSCI in vivo, as it is in spermatocytes cultured ex vivo (Fig. 1). This is a key question, because a previous in vivo study that used mouse models concluded that the maintenance of MSCI is independent of ATR and that MSCI is a stable and irreversible process once established[22]. To revisit this question, we treated wild-type mice with AZ20; we collected testes from mice 4 h after oral gavage with one dose of 50 mg/kg of AZ20 (Fig. 8a). We found that the γH2AX domain disappeared from the XY chromatin in H1T-positive mid-pachytene spermatocytes following treatment with AZ20, but not in testes from untreated mice (Fig. 8b), consistent with another independent study using the same condition[59]. To examine the maintenance of MSCI in this mouse model, we prepared chromosome spreads using cell suspensions. During the preparation of chromosome spreads, we continued to incubate cell suspensions with AZ20 for 30 min because we found that a 30-min incubation without inhibitor is sufficient to recover partial MSCI in culture (Fig. 5). We found that the γH2AX domain was attenuated in AZ20-treated mice, while, in controls, a short incubation with AZ20 on ice did not alter the γH2AX domain on the XY formed in vivo (Fig. 8c, d). Of note, the efficacy of in vivo administration of AZ20 was very high; no cell that had a clear γH2AX domain was detected (Fig. 8d). We combined Pol II immunostaining with γH2AX and SYCP3 staining and confirmed that disruption of the γH2AX domain by oral gavage with AZ20 led to Pol II loading onto the XY chromatin (Fig. 8e, f). Therefore, consistent with our culture experiments, we conclude that active DDR signaling is also required for the maintenance of MSCI in vivo.

## Discussion

In this study, we have established a system to study MSCI ex vivo, and have performed in vivo experiments in mice, to show that ATR-mediated DDR signaling actively governs MSCI during meiotic prophase I and functions as the primary mechanism of chromosome regulation on the XY. The ATR kinase is critical for both the mitotic and

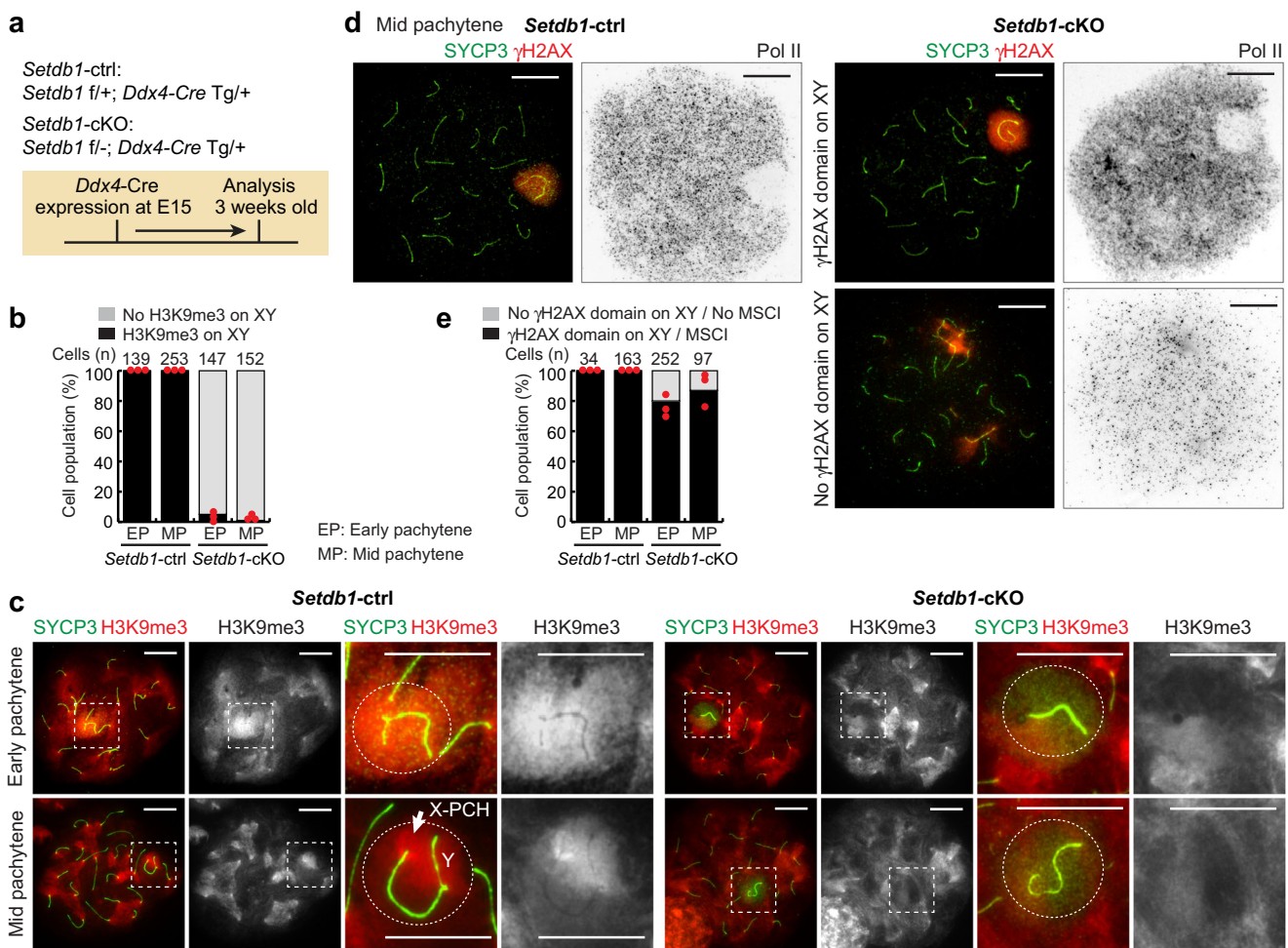

**Fig. 7 | SETDB1-dependent H3K9me3 is dispensable for the initiation of MSCI.**
**a** Schematic of the strategy for generating *Setdb1*-cKO mice. **b** Evaluation of the efficiency of SETDB1 deletion shown as the mean and individual values for 3 independent samples. **c** Chromosome spreads of early and mid pachytene spermatocytes, distinguished by the absence (early) or the presence (mid) of H1T, respectively (not shown), immunostained with antibodies raised against SYCP3 and H3K9me3. XY chromosomes are indicated with dashed squares and are magnified in the panels to the right. Putative regions of XY chromatin are indicated by dashed circles. **d** Chromosome spreads of mid pachytene spermatocytes immunostained with antibodies raised against SYCP3, γH2AX, and Pol II. **e** A population of pachytene spermatocytes with a normal γH2AX domain and normal MSCI, as defined by γH2AX and Pol II, respectively. The mean and individual values for 3 independent samples are shown. EP early pachytene, MP mid pachytene, X-PCH X-pericentromeric heterochromatin. Scale bars: 10 μm. Source data are provided as a Source Data file.

meiotic cell cycle, especially in regulating the checkpoint machinery, DNA replication, and recombination[19–21]. A recent study with AZ20, which was originally developed for chemotherapy, used an in vivo mouse model and a testicular organ culture model to show that ATR is required to complete meiotic recombination[20]. Another study used AZ20 for phosphoproteomics of ATR signaling in mouse testes using an in vivo mouse model[59]. In our study, we instead focused on treating mouse spermatocytes ex vivo with two different ATR inhibitors to specifically address the function of ATR in MSCI.

Here we find the unexpected result that, as part of its regulation by the DDR, MSCI is a reversible process. MSCI was previously thought to be a stable and irreversible process once the silencing machinery induces gene silencing downstream of the DDR[22,28], similar to how the silencing machinery is recruited to induce heritable silencing downstream of *Xist* non-coding RNA during female X-inactivation[1–4]. We demonstrate the reversibility of MSCI in two different ways: (1) by the ability of ATRi to suppress both the γH2AX domain and revert MSCI after they had already been established, and (2) by the recovery of these features upon release from treatment with ATRi. Thus, our study yields a regulatory mechanism for chromosome-wide silencing based on an active signaling process.

Our data also suggest that DDR signaling continuously takes place from the XY axes to the XY chromatin to maintain the γH2AX domain throughout the pachytene stage. This is interesting in the context of the model we previously proposed that MSCI is initiated by two genetically separable events[14]. In the first step, occurring on the axes of the XY chromosomes, ATR-dependent phosphorylation generates γH2AX. This, in turn, attracts the γH2AX binding partner MDC1. According to this model, MDC1 then facilitates the spread of γH2AX to chromatin loops of the XY in an ATR-dependent manner to eventually encompass the entirety of the XY chromosomes. Results shown here support the notion that such a regulatory mechanism actively maintains MSCI. In particular, γH2AX is reestablished and, simultaneously, Pol II is re-excluded from the chromatin adjacent to the axes of the XY in spermatocytes at early points of recovery from inhibition of DDR signaling by ATRi. Also consistent with our model, at longer time points of recovery, γH2AX is eventually reestablished, and Pol II was excluded, over the entirety of the XY chromosomes after ex vivo treatment of spermatocytes with ATRi.

In meiosis, regulation of the DDR is a key mechanism to monitor chromosome synapsis[60,61], and MSCI functions as a checkpoint by sequestering DDR proteins to sex chromosomes, thereby allowing

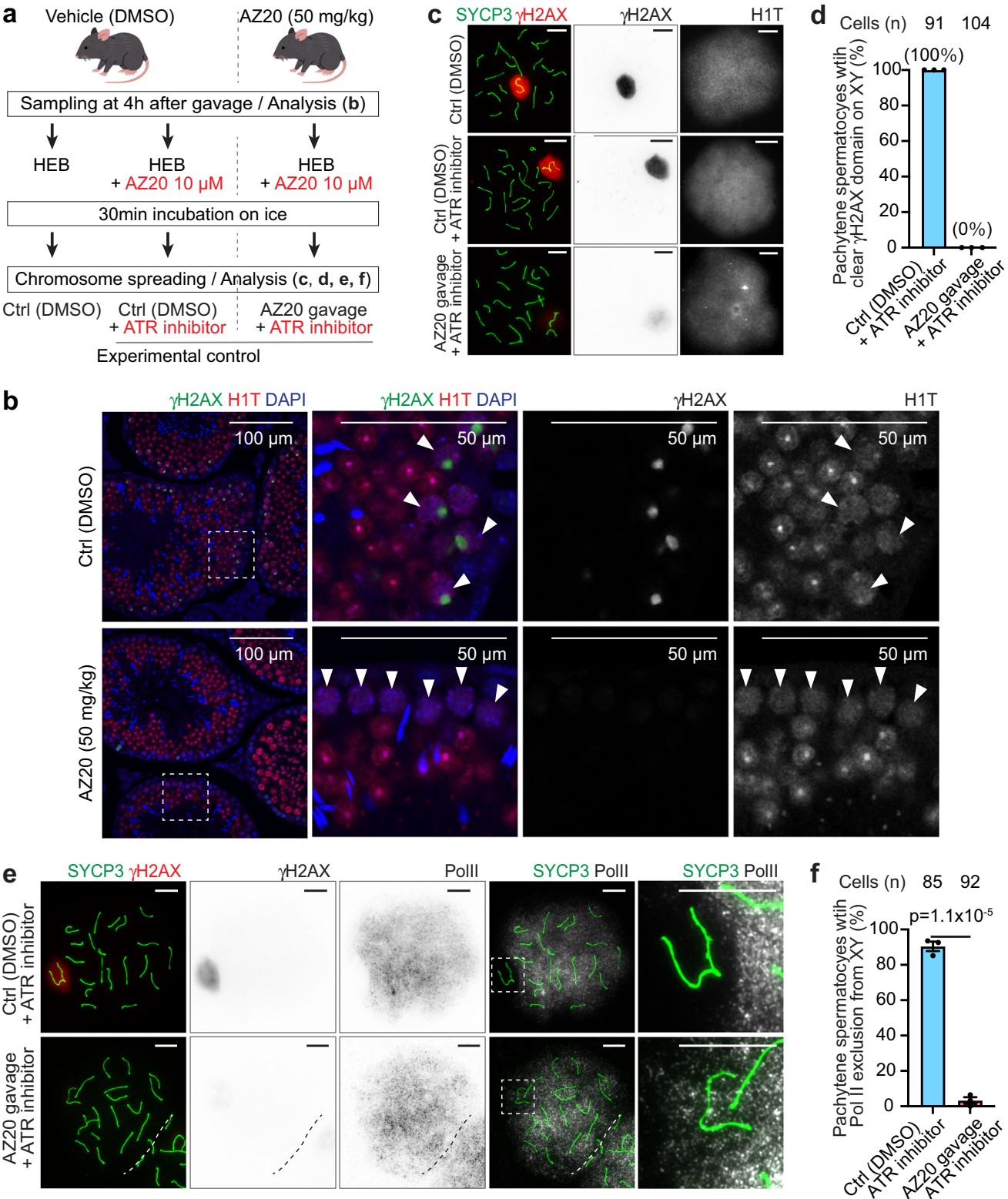

subsequent XY body formation and stage progression[44]. Thus, it is reasonable to postulate that the XY body functions as an active signaling hub centered, at least in part, on ATR, which controls male meiosis by integrating the checkpoint machinery with gene silencing. Given the multitude of functions of ATR outlined above, the XY body may serve as a signaling hub where ATR acts to coordinate various processes that are necessary for the successful completion of meiosis. Additionally, because γH2AX cannot be sustained when ATR is inhibited, and because ATR-dependent MSCI is reversible, these results may suggest that a sort of equilibrium exists between phosphorylation by ATR and dephosphorylation by a hitherto unidentified protein phosphatase.

Another important aspect of our study is that it clarifies, in contrast to reversible DDR signaling, that certain histone PTMs (H3K9me3 and H3K27ac) act downstream as a static layer of modifications on the XY chromatin. A previous RNA-seq analysis reported that sex

**Fig. 8 | Active DDR signaling is required for the maintenance of MSCI in spermatogenesis in vivo. a** Schematic of the experimental design. See Materials and Methods for more details. HEB hypotonic extraction buffer. **b** Testis sections immunostained with antibodies raised against γH2AX and H1T. Representative images of 3 independent experiments are shown. Dashed squares are magnified in the panels to the right. White arrowheads indicate H1T-positive spermatocytes. **c** Chromosome spreads of mid-late pachytene spermatocytes immunostained with antibodies raised against SYCP3, γH2AX, and H1T. **d** Quantification of mid-late pachytene spermatocytes with a clear γH2AX domain on the XY in chromosome spreads shown as the mean ± s.e.m. for 3 independent experiments. Mid-late spermatocytes with a clear γH2AX domain on the XY were not found in AZ20 gavage plus ATR inhibitor, while all control spermatocytes (DMSO plus ATR inhibitor) presented a clear γH2AX domain on the XY. **e** Chromosome spreads of mid-late pachytene spermatocytes immunostained with antibodies raised against SYCP3, γH2AX, and Pol II. XY chromosomes are indicated with dashed squares and are magnified in the panels to the right. **f** Quantification of mid-late pachytene spermatocytes with normal MSCI defined by Pol II exclusion shown as the mean ± s.e.m. for 3 independent experiments. Two-tailed unpaired t-test. Total numbers of analyzed nuclei are indicated in the panels (**d**, **f**). Scale bars: 10 μm unless otherwise noted in the panels. Source data are provided as a Source Data file. The copyright of the mouse illustration is attributed to Takashi Mifune (https://www.irasutoya.com/). All rights reserved.

chromosome-linked genes are upregulated in spermatocytes with conditional deletion of *Setdb1*; *Setdb1*-cKO pachytene spermatocytes exhibited 3.3- and 3.7-fold upregulation of X and Y-linked genes, respectively, as compared to their controls[28]. We cannot directly compare that study to our RNA-seq analysis, which shows massive upregulation of X and Y-linked genes (82.7- and 64.7-fold increases in mean expression from the X and Y chromosomes, respectively) in isolated pachytene spermatocytes after treatment with ATRi (Fig. 2b), due to various technical differences. However, differences of more than an order of magnitude in our results support the notion that the DDR is the direct mechanism for MSCI initiation and maintenance.

Still, our results suggest that H3K9me3 could possibly have a role in bolstering DDR-dependent MSCI. It should be noted that in our cytological experiments, we only evaluated MSCI in the *Setdb1*-cKO pachytene spermatocytes that completed autosomal synapsis. One possibility could be that MSCI is indirectly disrupted in *Setdb1*-cKO pachytene spermatocytes that have autosomal asynapsis. Nevertheless, the exact function of H3K9me3 remains a mystery, in part because the distribution of H3K9me3 is dynamic and temporally disappears from X-chromatin in mid-pachytene spermatocytes (presumably due to histone replacement)[32]. Additionally, H3K27ac is associated with escape gene activation on the sex chromosomes[49]; this also supports the notion that certain histone PTMs provide an additional layer of gene regulation on top of DDR-mediated primary silencing. In line with their functions as heritable epigenetic modifications through cell divisions, these irreversible histone PTMs, which are established on XY chromatin in meiosis, are maintained into postmeiotic spermatids[32,39,62].

Finally, another intriguing feature, driven by active DDR signaling, could be the droplet-like biophysical nature of XY chromatin, explained by phase separation[8,63]. Going forward, it will be critical to determine the link between phase separation and gene silencing. Our system to study MSCI ex vivo could be highly useful for addressing these questions in the future. However, we acknowledge the possible limitation of our culture method in that spermatocytes do not have a direct association with other cells, such as Sertoli cells, and are not interconnected with other spermatocytes in large syncytia. The faithful ex vivo recapitulation of the entire male meiotic process remains a significant challenge in the field. In conclusion, the concept of active DDR signaling as a central regulator of MSCI should provide a foundation for understanding phase separation and other key aspects of male meiosis and germ cell development.

## Methods
### Animals
Mice were maintained on a 12:12 light:dark cycle in a temperature and humidity-controlled vivarium (22 ± 2 °C; 40–50% humidity) with free access to food and water in a pathogen-free animal care facility. All experimental work was approved by the Institutional Animal Care and Use Committee under protocol No. 21931 at UC Davis. At least three independent experiments using different mice were performed for all culture experiments, except for RNA-seq which was examined with two biological replicates. In each experiment, testicular cells were isolated from a mouse, then split for treatment and control conditions. Three independent mouse pairs were used for in vivo analyses.

For culture experiments and AZ20 gavage treatments, mature male C57BL/6 (hereafter described as B6) mice greater than 6 weeks of age were used. To generate *Setdb1*-cKO (*Setdb1*^flox/flox^: *Ddx4-Cre*^Tg/+^) mice, a male carrying the *Setdb1*^flox/+ 55^: *Ddx4-Cre*^Tg/+ 56^ alleles was crossed with a female carrying the *Setdb1*^flox/flox^ allele. Males carrying the *Setdb1*^flox/+^; *Ddx4-Cre*^Tg/+^ alleles obtained from the same litter were used as littermate controls. The control and *Setdb1*cKO mice greater than 3 weeks of age were harvested and analyzed.

### Preparation of cell cultures
We modified a previously established method for short-term culture[29], which utilized isolated pachytene spermatocytes for culture. Because H1T-positive spermatocytes can be selectively analyzed by microscopy after culture, whole testicular cell suspensions were instead used for culture in our study. We cultured whole testicular cell suspensions using the same conditions described in the initial short-term culture method[29]. Briefly, testes collected from mature adult B6 mice were placed on a Petri dish, then the tunica albuginea was removed. The decapsulated testes were transferred into 2 mL of Enriched Krebs-Ringer Bicarbonate medium prepared as described[29] in a 15 mL tube and were incubated for 20 min at 35 °C with 1 mg collagenase (Washington Biochemical Corporation, CLS1) and 20 μg DNase (Sigma, D5025-150KU) to remove interstitial cells. The supernatant enriched with interstitial cells was removed after a pulse of centrifugation, then the remaining seminiferous tubules were digested by incubating in dissociation buffer (100 μg DNase and 4 mg collagenase in 2 mL of TrypLE™ Express (Gibco, 12604021)) for 20 min at 35 °C. The single-cell suspension that was obtained was then washed 3 times with Minimum Essential Medium alpha (MEMα) (Gibco, 12000-022) prepared as described[29], followed by centrifugation at 100 g, 5 min at 4 °C. The cell suspension was resuspended in MEMα and filtered using a 35 μm mesh (Falcon, 352235). Cell number in the suspension was counted using a TC20 Automated Cell Counter (Bio-Rad) for dilution to $2.5 \times 10^6$ cells/mL in MEMα containing DMSO (control culture media) or ATR inhibitors, seeded on a 4-well culture plate (Thermo Fisher Nunc, 055047), then incubated for 24 h at 32 °C with 5% $CO_2$. Two commercial ATR inhibitors were used: AZ20 (Sigma, SML1328) at 5 or 10 μM in culture medium (5 mM and 10 mM stocks were dissolved in DMSO) and AZD6738 (Selleck Chemicals) at 2 or 5 μM in culture medium (2 mM and 5 mM stocks were dissolved in DMSO). After 24 h culture, the viability of spermatocytes was measured using a TC20 Automated Cell counter with gating for a cell size of 10–15 μm.

For recovery experiments, both control and AZ20-treated cells were collected after 24 h incubation by centrifugation at 100 g for 5 min at room temperature (RT). Next, the cells were washed twice by suspending in the control culture media containing DMSO and centrifuging at 100 g for 5 min at RT. The cell pellets that were obtained were then resuspended in 1 mL of control culture medium, and the

cells were incubated for 30 min or 3 h at 32 °C with 5% $CO_2$ air (the same condition as 24 h culture with ATRi).

### ATR inhibitor treatment by gavage

ATR inhibitor treatment by gavage was performed as described[20,59]. Briefly, the AZ20 compound (MedChemExpress) was dissolved in 10% DMSO/40% propylene glycol/50% water and was administered orally to adult B6 males at a single dose of 50 mg/kg. Control animals were administrated the same volume of vehicle solution. Animals were sacrificed 4 h after the treatment.

### Preparation of meiotic chromosome spreads

**In vivo samples.** Meiotic chromosome spreads from testes were prepared essentially as described[64], with some modifications. Testes were excised, detunicated, and placed in 1× phosphate-buffered saline (PBS). Seminiferous tubules were dissociated from whole testes, and approximately one-quarter of an adult testis was used for meiotic chromosome spreads. Seminiferous tubules were transferred to one well of a four-well dish (Thermo Fisher Nunc, 055047) containing 1 mL of cold PBS kept on ice. Seminiferous tubules were then gently unraveled into small clumps with fine-point tweezers; care was taken not to tear or mince the tubules. The clumps of seminiferous tubules were subsequently transferred to the second and third wells of 1 mL PBS for additional unraveling before transfer to the fourth well, which contained 1 mL hypotonic extraction buffer [HEB: 30 mM Tris base, 17 mM trisodium citrate, 5 mM ethylenediaminetetraacetic acid (EDTA), 50 mM sucrose, 5 mM dithiothreitol (DTT), and 1× cOmplete Protease Inhibitor Cocktail (Sigma, 11836145001), pH. 8.2]. Once there, fine-point tweezers were used to carefully expose the surface area of tubules to HEB. The seminiferous tubules were incubated in HEB on ice for 30 min with gentle stirring every 10 min. After incubation, seminiferous tubules were mashed using a disposable surgical blade in 60 µL of 100 mM sucrose on a plain, uncharged microscope slide (Gold Seal: ThermoFisher Scientific, 3010–002). After approximately 15–25 mashes, a semi-translucent cell suspension was formed. An additional 120 µL of sucrose (100 mM) was added to the suspension, and the suspension was mixed via gentle pipetting up and down several times. The diluted cell suspension was applied to positively charged slides (Probe On Plus: ThermoFisher Scientific, 22-230-900) in 30 µL volumes; before application of the suspension, the slides had been incubated in chilled fixation solution (2% paraformaldehyde [PFA], 0.1% Triton X-100, and 0.02% sodium monododecyl sulfate, adjusted to pH 9.2 with sodium borate buffer) for a minimum of 2 min. After applying the cell suspension/sucrose mixture, the slide was slowly, gently tilted up and down at slight angles (<10°) to mix the cell suspension/sucrose mixture with the remaining fixation solution. The slides were placed in "humid chambers" (closed pipet tip boxes filled to approximately two-thirds volume with water) at RT for a minimum of 1 h (maximum overnight). Then, the slides were washed in a low-concentration surfactant, 0.4% Photo-Flo 200 (Kodak, 146–4510), at RT two times, 2 min per wash. Slides were dried completely at RT (~30 min) before staining or storage in slide boxes at −80 °C.

Of note, AZ20 (final 10 µM in the solutions) was added to PBS, HEB, and 100 mM sucrose to prevent diffusion of orally administrated AZ20. No effect on γH2AX domain formation during incubation with additional AZ20 was confirmed by comparing control samples with/without additional AZ20 in PBS, HEB, and 100 mM sucrose (see Fig. 8a, c).

**Cultured cell samples.** Cultured cells collected from each well (ideally $2.5 \times 10^6$ cells) were centrifuged at $600 g$, 5 min at 4 °C, resuspended in 1 mL HEB with DMSO or ATR inhibitors at the same concentration as utilized in the culture medium, then incubated for 5 min on ice. After gentle pipetting, a 30 µL aliquot of the cell suspension in HEB was directly applied to positively charged slides coated by chilled fixation solution as described above. Subsequent processing was the same as for in vivo samples.

**Immunostaining of meiotic chromosome spreads.** For immunostaining experiments, the chromosome spreads were incubated in PBS containing 0.1% Tween 20 (PBST) for 5–30 min before blocking in antibody dilution buffer [PBST containing 0.15% bovine serum albumin (BSA)] for an additional 30–60 min. Primary and secondary antibodies (described below) were diluted in antibody dilution buffer. Then, chromosome spreads were coated with 100 µL of the antibody/antibody dilution buffer solution, gently covered with Parafilm (Parafilm M All-Purpose Laboratory Film, Bemis Company, Inc.), and stored in humid chambers at 4 °C for a minimum period of 6 h to a maximum timespan of overnight (-15 h). This study made use of the following primary antibodies at the following dilutions [format: host anti-protein (source or company with product/catalog number if applicable), dilution]: rabbit anti-SYCP3 (Novus, NB300–232), 1/500; mouse anti-SYCP3 (Abcam, ab97672), 1/5000; goat anti-SYCP3 (R&D Systems, AF3750), 1/200; mouse anti-H2AX-pS139 (γH2AX: Millipore, 05–636), 1/5000; mouse anti-H2AX-pS139 (γH2AX) conjugated to Alexa 647 fluorophore (Millipore, 05–636-AF647), 1/2000; rabbit anti-TOPBP1 (a gift from Dr. Junjie Chen[65]), 1/2000; rabbit anti-BRCA1 (generated in the Namekawa Lab[14]), 1/500; sheep anti-MDC1 (Bio-Rad, AHP799), 1/500; guinea pig anti-H1T (a gift from Dr. Mary Ann Handel[33]), 1/2000; rabbit anti-ATR (Millipore, PC538), 1/2000; mouse anti-Pol II (Santa Cruz, sc-56767), 1/100; rabbit anti-H3K27ac (Active Motif, 39133), 1/2000; mouse anti-SUMO1 (Invitrogen, 33-2400), 1/200; rabbit anti-Ubiquitin (Abcam, ab19247), 1/200; mouse-H3K9me3 (Abcam, ab8898), 1/200; rabbit anti-SCML2 (generated in the Namekawa Lab[53]), 1/500; rabbit anti-SETDB1 (Proteintech, 11231-1-AP), 1/200. After incubation with primary antibodies, slides were washed three times in PBST, 5 min per wash. Then, the slides were incubated with the appropriate secondary antibodies conjugated to Alexa 488, 555, and/or 647 fluorophores (ThermoFisher Scientific). All secondary antibodies were diluted 1/500 in antibody dilution buffer. Slides were coated with 100 µL of the antibody/antibody dilution buffer solution; then, they were gently covered with Parafilm for 1 h incubation at RT in humid chambers in darkness. After slides were washed three times in PBST in darkness, 5 min per wash, they were counterstained with the DNA-binding chemical 4′,6-diamidino-2-phenylindole (DAPI; Sigma, D9542–5MG) diluted to 1 µg/mL concentration in PBS. Finally, slides were mounted using 20 µL undiluted ProLong Gold Antifade Mountant (ThermoFisher Scientific, P36930). Slides were either imaged immediately or stored at 4 °C in darkness. For long-term storage, stained slides were kept at 4 °C in darkness.

Images were obtained with an ECLIPSE Ti-2 microscope (Nikon) equipped with an ORCA-Fusion sCMOS camera (Hamamatsu) and a 60× CFI Apochromat TIRF oil immersion objective NA 1.4 (Nikon) and were processed using NIS-Elements Basic Research (Nikon), Photoshop (Adobe), and Illustrator (Adobe) software.

**Judgement of pachytene substages.** Histone variant H1T starts to express from the mid-pachytene stage in mice[66], and the pachytene duration is approximately 160 h (7 days) in mouse spermatogenesis[32]. To focus on maintenance mechanisms of DDR and MSCI, we analyzed H1T-positive mid-late pachytene spermatocytes because these spermatocytes were already in the pachytene stage, carrying a fully established γH2AX domain on the XY at the beginning of the ATR inhibitor exposure (Supplementary Fig. 1). For additional judgments of pachytene substages, we applied a criterion based on the morphology of the chromosome axis visualized by the SYCP3 antibody, as previously described[67].

## Preparation of sample slides for RNA fluorescence in situ hybridization (FISH) and combined immunofluorescence

**Cultured cell samples.** Slide preparation from a single cell suspension was performed as described previously[68] with some modifications and was used for the analysis of cultured cells. Cultured cells collected from each well (approximately $2.5 \times 10^6$ cells per well) were centrifuged at $600\,g$, 5 min at 4 °C, and resuspended in 600 μL cold CSK buffer (100 mM NaCl, 300 mM sucrose, 10 mM PIPES, and 3 mM MgCl$_2$, the pH adjusted to 6.8 with 1 M NaOH) + 0.5% Triton X-100 with DMSO or ATR inhibitors at the same concentration as the culture medium. 100 μL of cell suspension was placed on each Superfrost /Plus Microscope Slide (Fisher, 12-550-15) and incubated for 15 min at 4 °C. CSK buffer was drained from the slides by placing a paper towel at one end of the slides, and then the slides were gently tilted. The cells were fixed by gently adding ice-cold 100 μL of 4% PFA to the slides; the slides were incubated for 10 min at 4 °C. The slides were gently tilted to drain the 4% PFA. Then, using a Coplin jar containing PBS, the slides were washed for 5 min at RT. The slides were dehydrated in a Coplin jar containing 70%, 80%, and then 100% ethanol for 2 min each and air-dried completely. The slides were stored at −80 °C in a slide box.

**In vivo samples.** Slide preparation from mouse testicular tubules for RNA FISH was performed as described previously[69,70] with some modifications, and these slides were used for the analysis of *Setdb1*-cKO and *Setdb1*-ctrl testes. This method was developed and optimized for RNA FISH to detect nascent transcription by removing cytoplasmic backgrounds[69,70]. Testes were excised, detunicated, and placed in 1× PBS. Seminiferous tubules were dissociated from whole testes, and approximately one-quarter of an adult testis was transferred into one well of a 4-well dish containing 500 μL of CSK buffer +0.5% Triton X-100 on ice and incubated for 6 min. After incubation, all the tubules were transferred to another well of a 4-well dish containing 4% PFA-PBS solution and incubated for 10 min at RT. All the tubules were then transferred into 30 μL of PBS on a non-charged glass slide. Using the tips of two forceps, the tubules were torn into pieces. The tubules were chopped in a horizontal direction by clipping the tubules between the tips of two forceps and pulling the forceps horizontally. This step was continued for approximately 10 sec. The suspension was mixed by pipetting using a P20 pipette. The suspension was transferred to a microcentrifuge tube and diluted with PBS to approximately 1.3 mL. 100 μL each of suspension was applied to twelve cytospin chambers, and cytospin was performed at $30 \times g$ (2000 g for 10 min at RT. After cytospin, slides were dried on a lab bench for a few minutes at RT. Using a Coplin jar containing PBS, the slides were washed for 5 min at RT. The slides were dehydrated using serial treatment with 70%, 80%, and then 100% ethanol for 2 min each and air-dried completely. The slides were stored at −80 °C in a slide box. For Cot-1 RNA FISH, slides were also prepared from *Setdb1*-cKO and *Setdb1*-ctrl testes using the single cell suspension method described above, and consistent results were obtained between these two slide preparation methods.

**Gene-specific RNA FISH and combined immunostaining.** Gene-specific RNA FISH probes to detect nascent transcripts of *Utx* and *Lamp2* genes (*Utx* DNA probe conjugated with FITC and *Lamp2* DNA probe conjugated with Cy3) were prepared as described previously[37,69]. For this purpose, DNA probes to hybridize target nascent RNAs were generated by labeling via nick translation of template genomic DNAs. These DNA probes do not hybridize to genomic DNA without denaturation, thereby detecting target nascent transcripts specifically. Gene-specific RNA FISH was performed as described[37,69], with some modifications. The slides were dehydrated by serial treatment with 70%, 80%, and 100% (vol/vol) ethanol in Coplin jars for 2 min each and air-dried completely. 20 μL/slide of

hybridization buffer (2× SSC, 10% dextran sulfate, 1 mg/mL BSA, 1 mM vanadyl ribonucleoside, and 50% formamide) containing probes (100 ng/μL each) were denatured at 80 °C for 10 min and preannealed [with non-specific competitor DNAs and RNAs (herring sperm DNA, mouse Cot-1 DNA, and yeast transfer RNAs in the probe solution)] at 37 °C for 10 min using a PCR machine. The preannealed probe solution was applied to the slides and carefully covered with cover glasses and sealed with Elmer's rubber cement. Incubation was performed in a humidified chamber at 37 °C overnight (16 h). The next day, cover glasses were removed, and the slides were washed twice in Coplin jars containing 50% (vol/vol) formamide in 2 × SSC for 5 min at 37 °C and further washed twice in Coplin jars containing 2× SSC for 5 min at 37 °C. The slides were washed in a Coplin jar with PBST for 5 min at RT. Then, immunostaining of HORMAD1 was performed on the slides to detect unsynaped XY axes. The slides were incubated with 100 μL of a primary antibody solution (1/200 dilution: rabbit anti-HORMAD1, a gift from Dr. Attila Toth[58] in PBST containing 0.15% BSA) at RT for 2 h in humid chambers in darkness. After incubation with the primary antibody, the slides were washed three times in Coplin jars containing PBST (5 min per wash). Then, the slides were incubated with 100 μL of secondary antibody solution (1/500 dilution of a secondary antibody conjugated to Alexa 647 fluorophore (ThermoFisher Scientific) in PBST containing 0.15% BSA) for 1 h incubation in humid chambers in darkness. After slides were washed three times in PBST in darkness, 5 min per wash, they were counterstained with DAPI diluted to a concentration of 1 μg/mL in PBS. Finally, slides were mounted using 10 μL undiluted ProLong Gold Antifade Mountant (ThermoFisher Scientific, P36930). Slides were either imaged immediately or stored at 4 °C in darkness.

**Cot-1 RNA FISH and combined immunostaining.** Cot-1 RNA FISH probe conjugated with Cy3 to detect nascent nuclear transcripts was prepared as described previously[39,69]. The Cot-1 DNA probe to hybridize global nascent RNAs was generated by labeling via random priming of template Cot-1 DNA (Invitrogen, 18440016). Cot-1 RNA FISH was performed as described[39,69] with some modifications. The slides were dehydrated by serial treatment with 70%, 80%, and 100% (vol/vol) ethanol in Coplin jars for 2 min each and air-dried completely. 10 μL/slide of hybridization buffer (2× SSC, 10% dextran sulfate, 1 mg/mL BSA, 1 mM vanadyl ribonucleoside, and 50% formamide) containing Cot-1 probes (100 ng/μL) was denatured at 80 °C for 10 min and preannealed (with non-specific competitor DNA: herring sperm DNA in the probe solution) at 37 °C for 10 min using a PCR machine. The preannealed probe solution was applied on the slides and carefully covered with cover glasses. Incubation was in an humidified chamber at 37 °C overnight (16 h). The next day, the cover glasses were removed, and the slides were washed twice in Coplin jars containing 50% (vol/vol) formamide in 2× SSC for 5 min at 37 °C and further washed twice in Coplin jars containing 2× SSC for 5 min at 37 °C. The slides were washed in a Coplin jar with PBST for 5 min at RT. Then, immunostaining of HORMAD1 was performed as described above, and a secondary antibody conjugated to Alexa 488 fluorophore was used.

## Preparation of RNA-seq samples and data analysis

**Fluorescence-activated cell sorting (FACS) followed by short-term spermatocyte culture.** We followed a published protocol designed to isolate pachytene spermatocytes from adult mice[71]. Briefly, the testicular cell suspensions from B6 mice (23 weeks of age) were prepared by performing the preparation steps of the short-term culture described above. The cells were suspended in 3 mL of PBS containing 2% FBS (FACS buffer) after filtration with 35 μm mesh, instead of MEMα, and were then stained with 6 μL of a DNA binding-dye DyeCycle Violet (Thermo Fisher, V35003), for 1 h, at 34 °C. After filtration with 35 μm mesh and the addition of 5 μg of DNase I,

testicular cells in the FACS buffer were sorted by a Sony SH800S cell sorter in which a 130 μm tip was installed. We followed a gating parameter in the published protocol[71]. A fraction enriched with pachytene spermatocytes was collected into a 15 mL tube containing 2 mL of 100% FBS. The cells were washed twice by MEMα. A portion of the isolated pachytene spermatocytes was used to confirm purity. For this purpose, 2 μL of cell suspension was taken and incubated in 60 μL of HEB for 5 min on ice. Purities of pachytene spermatocytes were confirmed by staining meiotic chromosome spreads as described above with antibodies raised against SYCP3, γH2AX, and H1T (Supplementary Fig. 4a, c). The isolated pachytene spermatocytes were diluted to $2.5 \times 10^6$ cells/mL and incubated with DMSO or 10 μM AZ20 for 24 h at 32 °C with 5% $CO_2$.

**Spermatocyte collection.** After culturing for 24 h, the spermatocytes were collected into 1.5 mL tubes and washed with PBS, followed by centrifugation at 100 g, 5 min at 4 °C, three times. Cell number after final washing was calculated using a TC20 Automated Cell Counter (Bio-Rad), and 15,000 cells from each experiment were used for RNA-seq analysis. To confirm attenuated DDR signaling on XY chromatin after treatment with AZ20, ~30 μL of aliquots of cultured spermatocytes were used for meiotic chromosome spreads and then immunostained with antibodies raised against SYCP3 and γH2AX.

**RNA-seq library generation and sequencing.** Total RNA was extracted from 15,000 cultured spermatocytes in each experimental group using the RNeasy Plus Micro Kit (QIAGEN, 74034) following the manufacturer's instructions. RNA-seq library preparation was performed with NEBNext® Single Cell/Low Input RNA Library Prep Kit for Illumina® (NEB, E6420S) following the manufacturer's instructions. Prepared RNA-seq libraries were sequenced on the HiSeq X-ten system (Illumina) with paired-ended 150-bp reads. Two independent biological replicates were generated for the RNA-seq library.

**RNA-seq data processing.** Raw RNA-seq reads were trimmed and filtered using trimmomatic[72] v0.39 [http://www.usadellab.org/cms/?page=trimmomatic] and then aligned to the mouse genome (GRCm38/mm10) using STAR[73] version 2.5.4b with --outFilterMultimapNmax 1 option for unique alignments. To identify differentially expressed genes between experimental groups, raw read counts files were generated using featureCounts[74] v2.0.1 based on the UCSC mouse refGene annotation. After quantification, a read count file was input to the DESeq2 package[75] (version. 1.36.0) in R, and DESeq2 was used for differential gene expression analyses. Differentially expressed genes were identified with cutoffs ≥2-fold change and a false discovery rate (FDR) values ($P_{adj}$; $p$-values adjusted for multiple testing using the Benjamini–Hochberg method) of <0.05 among expressed genes (TPM > 1 in either the Ctrl or the AZ20-treatment). TPM values for genes were calculated from read counts. Statistical significance for autosome and sex chromosome comparisons were determined using the Wilcoxon rank sum test. The average fold-changes for autosomes and sex chromosomes were calculated from the average TPM per chromosome for Ctrl and AZ20-treatment condition. Genes with TPM > 1 in the Ctrl or the AZ20-treatment were defined as expressed genes and used in the analysis. To compare biological replicates, Pearson correlation coefficients were calculated using the corrplot package (version. 0.92) in R. Heatmaps of Pearson correlation were produced using ggplot2 (3.3.6) in R.

**Histology and immunostaining**
For the preparation of testis paraffin blocks, obtained testes were fixed with 4% paraformaldehyde and 0.1% Triton X-100 in PBS at 4 °C overnight. Testes were dehydrated and embedded in paraffin. For histological analyses, 6 μm-thick paraffin sections were deparaffinized and stained with hematoxylin and eosin. TUNEL assays were performed on 6 μm-thick paraffin sections using an In Situ Cell Death Detection Kit (Roche, 11684795910) per the instructions in the manual. For immunostaining, 6 μm-thick paraffin sections were autoclaved in Target Retrieval Solution, Citrate pH 6.1 (DAKO, S-1700) at 121 °C and 100 kPa (15 psi) for 10 min. The sections were blocked with Blocking One Histo (Nacalai USA, 06349–64) at RT for 10 min; the sections were then incubated with primary antibodies diluted in PBST at 4 °C overnight. The following antibodies were used at the following dilutions [format: host anti-protein (source or company with product/catalog number), dilution]: mouse anti-H2AX-pS139 (γH2AX) conjugated to Alexa 647 fluorophore (Millipore, 05–636-AF647), 1/2000; guinea pig anti-H1T (a gift from Dr. Mary Ann Handel)[33], 1/2000. The resulting signals were detected with secondary antibodies conjugated to Alexa 555 (ThermoFisher Scientific), diluted 1/1000 in PBST, and incubated at RT for 1 h. Sections were counterstained with DAPI as described above.

Images were obtained with a Zeiss LSM800 confocal microscope equipped with an Axiocam 506 mono (Zeiss) and a C-Apochromat 40x/1.2 water lens (Zeiss), and were processed using Fiji (NIH), Photoshop (Adobe), and Illustrator (Adobe) software.

**Quantification and statistical analysis**
The details for statistical analyses performed in this study are described in relevant portions of the Results section and figure legends. Sample sizes used for analyses are described in relevant portions of the Results section and figure legends. In predetermining sample sizes, we sought to analyze a minimum of three independent samples. No data were excluded from analyses. The experiments were not randomized, and investigators were not blinded to allocation during experiments and outcome assessment. Measurements were recorded in Excel (Microsoft) and Prism 9.0 (GraphPad). Statistical tests were performed with Prism 9.0 (GraphPad).

**Reporting summary**
Further information on research design is available in the Nature Research Reporting Summary linked to this article.

## Data availability
RNA-seq data reported in this study were deposited to the Gene Expression Omnibus with the GEO series accession number GSE211519). The raw data of quantifications presented in main figures and supplementary figures are provided as Source data with this paper. Source data are provided with this paper.

## Code availability
Source code for all software and tools used in this study with documentation, examples, and additional information, is available at the following URLs: trimmomatic (http://www.usadellab.org/cms/?page=trimmomatic), STAR (https://github.com/alexdobin/STAR), featureCounts (http://subread.sourceforge.net), DESeq2 (https://bioconductor.org/packages/release/bioc/html/DESeq2.html), corrplot (https://github.com/taiyun/corrplot) and ggplot2 (https://github.com/tidyverse/ggplot2).

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

## Acknowledgements

We thank members of the Namekawa laboratory for discussion and helpful comments regarding this manuscript. We also thank K.G. Alavattam for discussion, Y. Shinkai for sharing the *Setdb1*-floxed mouse line, N. Hunter for sharing the confocal microscope and discussion, J. Chen for providing the anti-TOPBP1 antibody, A. Toth for providing the anti-HORMAD1 antibody, and M.A. Handel for providing the anti-H1T antibody. Funding sources: KAKENHI #19H05743 to K.I.I.; NIH R01 GM134731 to P.R.A.; UC Davis startup fund, and NIH R01 GM098605 and R35 GM141085 to S.H.N.

## Author contributions

H.A. and S.H.N. designed the study. H.A. performed ex vivo and in vivo ATRi experiments, and Y.H.Y. performed RNA FISH experiments, *Setdb1*-cKO mice analyses and helped H.A. to analyze ex vivo experiments. Y.M. performed RNA-seq analysis. H.A., Y.H.Y., Y.M., K.I.I., P.R.A., and S.H.N. interpreted the results. H.A., P.R.A., and S.H.N. wrote the manuscript with critical feedback from Y.H.Y., Y.M., and K.I.I. H.A. and Y.H.Y. contributed equally. S.H.N. supervised the project.

## Competing interests
