## [Peer Review File · Nature Communications]

Active DNA damage response signaling initiates and maintains meiotic sex chromosome inactivationREVIEWER COMMENTS

Reviewer #1 (Remarks to the Author):

Abe et al. use an ex vivo system to address whether DNA damage response (DDR) is necessary for both initiation and maintenance of meiotic sex chromosome inactivation (MSCI). Using an ATR inhibitor (AZD6738), they find ATR-dependent DDR is necessary for initiation of markers of MSCI (gammaH2AX, PolII exclusion from the XY chromosomes, etc). In addition, after removal of ATR inhibition, these markers are rapidly reestablished, indicating that ATR-dependent regulation of these markers is reversible. They follow up their cell culture experiments with validation in vivo in mice.

This manuscript is a clear, concise exploration of the role of ATR in regulating key factors involved in MSCI. The approaches are logical, the data are very consistent, and the narrative is clear and easy to follow. The follow up with in vivo experiments address concerns that might arise from ex vivo data that conflict with previous studies. The major concern is the authors assumption that gene silencing, and thus MSCI, is also reversed together with the markers of MSCI. This is a huge assumption and there is no data to support that the X is reactivated upon use of the ATR inhibitor. The manuscript would benefit from addressing this issue and modifications to figures, and the experimental approach.

Major:

1. As described above the current writing of the manuscript is that MSCI, and thus gene silencing, is reversible with an ATR inhibitor. This assumption is not supported by any data indicating that X-linked genes are reactivated upon treatment with an ATR inhibitor. The presence of Pol2 or histone PTMs, cytologically, is insufficient in assessing whether there is active gene transcription from the X chromosome. The authors should either rewrite the entire text, eliminating the impact on gene silencing or MSCI, or, preferably, demonstrate that X-linked genes are actively being transcribed after ATRi. To demonstrate X-linked genes are actively transcribed, the authors should detect chromosome-wide nascent transcription (e.g. via GRO-seq) or at a minimum with mRNA-seq combined with RNA-FISH of a few loci.

2. Figure 3b-d: This figure shows loss of reestablishment of MSCI after ATRi is stopped. However, these images alone are indistinguishable from those in which ATRi never occurred. Images from parallel cultures showing establishment first would be more convincing.

3. The authors frequently compare MSCI to X chromosome inactivation, in particular noting that MSCI occurs much more rapidly. This is perhaps not surprising, given that the canonical role of the DDR is to rapidly respond to DNA lesions, whereas X inactivation relies on the expression of an RNA from a single locus, which coats the chromosome starting a cascade of PTM events. The overt comparisons between these two pathways could be reduced to a single comment in the discussion.

Minor:

1. Figure 1a: The cartoon of the experimental approach should include the drug treatment timeline to clearly explain when ATRi starts and stops in order to analyze mid-late pachytene stage cells.

2. All figures: Representative images are provided to show effects of ATRi on localization of BRCA1, exclusion of XY from PolII machinery, etc. As in Figure 1, graphs depicting the quantifications of these data should be presented.

3. These findings will be of interest to the non-meiotic DDR community, who may not be familiar with MSCI. Thus, a more general introduction into DDR, or their findings in light of non-meiotic DDR pathways in the discussion, would be of greater value to the community. For example, it is not clear

what the initial DNA damage signal is that leads to ATR-dependent MSCI, and how factors such as Spo11, ATM, or RPA are involved.

4. The authors references could be more unbiased to their own laboratory's studies. For example, on lines 182+183 the authors reference two studies (Refs. #29 and #30) that found genes "escaping" post-meiotic transcriptional repression in spermatids. Both of these references are from the senior author's laboratory. However, a variety of studies, from multiple groups, have demonstrated sets of genes that "escape" post-meiotic repression. Or for example, "Silencing of unsynapsed meiotic chromosomes in the mouse" (Turner et al., 2005) is a landmark publication with direct relevance to these studies, establishing the lack of synapse as a mechanism for meiotic silencing, however it is not referenced once in the manuscript.

Reviewer #2 (Remarks to the Author):

In mammals, homologs that fail to synapse during meiosis are transcriptionally inactivated. This process of transcriptional inactivation or meiotic silencing, drives inactivation of the heterologous XY bivalent in male germ cells and is named meiotic sex chromosome inactivation [MSCI]. MSCI is hypothesized to work as a meiotic surveillance mechanism in meiosis. During the last decade, mouse genetic analysis has established that ATR (within the DDR) directs MSCI (Royo et al., 2013; PMID: 23824539). It has been established that ATR first regulates HORMAD1/2 phosphorylation and localization of BRCA1 and TOPBP1 at unsynapsed axes. In more advanced states, ATR transduce the signal through MDC1 and H2AFX. And finally, ATR catalyzes histone H2AFX phosphorylation (γ H2AX formation) and this promotes through TRIM28 the SETDB1-dependent deposition of H3K9me3, XY condensation, and gene silencing (Hirota et al., 2018; PMID: 30393076).

In this MS, Abe et al., make use of a pharmacological inhibition of ATR to study the known functions of ATR in this pathway during a short time window and in a reversible manner using a short-term spermatocyte culture protocol. They conclude that two layers of modification are operating in the MSCI pathway one reversible (γ H2AX) and one irreversible (H3K9me3). They conclude that active ATR is the primary mechanism of silencing in MSCI.

There are several criticisms to the MS by Abe et al.

-The authors state in the text and in the abstract that they have established a novel ex vivo system to study MSCI and demonstrate that active DDR signaling is required to initiate and maintain MSCI. The first point is that the short-term ex vivo cell culture they have used is not a novel system as it has been applied in lot of previous occasions since it was described by the Handel lab (PMID: 19685331) as a modification of the previous method (PMID: 3651542). If the method is indeed new, they should compare its performance in comparison with the already published. Two major issues must be considered in assessing the function of spermatocytes under these ex vivo conditions ; i) that they lack their surrounding Sertoli cells and are no longer interconnected with other spermatocytes in large syncytia and that as consequence ii) the culture / experimental setting cannot exceed 24h. This is not trivial and constitutes a serious limitation to the whole study. There is an alternative cell culture condition that allow longer culturing of seminiferous tubules in air-liquid interface. This alternative is not plausible since it would require experimental design of the whole work.

The other point is that it was already known that ATR is needed to promote MSCI. In this regard, the authors always refer to the analysis of the DDR pathway throughout the paper when they indeed only modify / alter / study the activity of ATR through its pharmacological inhibition making use mainly of the short-term culture of spermatocytes. The only genetic approach carried out by the authors in the MS (cKO of SETDB1) has already been analyzed previously making use of a similar transgenic CRE (Hirota et al., 2018; PMID: 30393076). The authors conclude that the presence of γ H2AX in the absence of H3K9Me3 leads to meiotic silencing. However, the results shown in figure 4C are not

demonstrative (there is no a clear exclusion of Pol II immunolabeling) and in order to raise such a conclusion several additional more robust molecular procedures should have been used to firmly proof this ascertain. By contrast, the RNAseq results by the Turner lab show active transcription of inactive genes in the spermatocytes of the cSETDB1KO encoded by X and Y chromosomes lacking H3K9me3 ((between 3.3 and 3.7 times, respectively) and also by direct in situ hybridization of sex-linked RNA probes in spermatocytes (PMID: 20360682, Figure 5) despite showing γ H2AX labelling in the XY axes and in their corresponding chromatin (sex body) (see Figure 3a in PMID: 30393076).

Taking as a whole, the study is mostly descriptive and is mostly based on chemical inhibition of ATR (with the obvious limitations) and lack functional and mechanistic analysis. This is a strong limitation together with the lack of novelty or reduced impact of most of the descriptions carried out in the MS.

Other comments:

The number of animals and cells analyzed should be specified for each experiment of the different figures. The results should always be quantified and statistically analyzed.

Reviewer #3 (Remarks to the Author):

In this manuscript of the Namekawa lab, the authors study the role of the DDR in the establishment and maintenance of the meiotic sex chromosome inactivation (MSCI). To do so, they developed an ex vivo system that allows them to analyze MSCI and prove that active DDR signaling is required to start and maintain MSCI in a dynamic and reversible process. In this work, the authors describe reversible- and irreversible-PTM modifications linked to MSCI and demonstrate that the DDR is responsible for silencing the sex chromosomes during meiosis independently of H3K9me3.

This work represents a significant advance in understanding how sex chromosomes are silenced during the meiotic prophase. Furthermore, it describes a novel ex vivo system that permits the dynamic study of mammalian meiotic prophase in a way that has not been done before and might be of great utility to overcome the intrinsic limitations of the study of mammalian meiosis. The logic behind each experiment is well presented. The quality of the data displayed is very high, and the presented data seem to support the conclusions raised by the authors. Nonetheless, I have several major and minor comments I would like the authors to address before publications:

Major comments:

-The main comment I have about this study is the use of immunostaining against the RNA polymerase II as the single readout of gene expression. I understand that this marker has been extensively used to indicate sex chromosome inactivation, but I do not think the absence of RNA pol II signal on the sex chromosomes is sufficient to conclude that MSCI works appropriately. In my opinion, the authors should also perform RNA-FISH against particular XY genes (e.g., Scml2, Zfx,...) to show that they are adequately silenced.

-I think the authors should make an effort to analyze the outcome of the experiments they perform more objectively. Only a minority of the figures (e.g., 1b, d, f) have some quantification of the data observed and statistical analysis of those. Although the images provided in most of the figures are clear, objective quantification of the results should be provided for each of the experiments performed (e.g., Figures 2, 3d, 4, 5, 7, S1b, S2, S3, S4, and S5).

Minor comments:

-Line 75: Please explain a little bit more the novelty of this ex vivo system and how it relates to previous culture systems used to study ATR function in meiosis (Pacheco et al. 2018; Sims et al. 2021)

-I would suggest to the authors to include the information displayed in supplemental figure 4 in Figure 5

-Line 216: I think authors should provide PAS-H stainings to show better the meiotic arrest occurring

in Setdb1-cKO and, if possible, perform a TUNEL assay on these sections.

-Line 217: I do not think the IF against Setdb1 is enough to show the reduction of Setdb1 expression on conditional mutants. I want to propose that the authors perform a western blot analysis of the whole testis protein extracts to certify the status of the conditional mutant.

-Figure 2: The shape of the XY chromosomes varies during meiotic prophase and can be used to stage spermatocytes (Page et al., 2012). As the pachytene stage progresses, the X chromosomes curls. Thus, early pachynema cells have long, straight X chromosomes, and mid/late pachynema cells have mostly curly X chromosomes. The X and Y chromosomes are displayed differently in the cartoons provided for control and AZ20-treated samples. While in the controls, the X is curled; in the treated samples, the X is straight. I wonder if this is drawn on purpose and if it reflects a different conformation of the XY chromosomes in these two populations.

-Figure 3b: In this case, I think it is very relevant to quantify the intensity of the gH2AX signal to know the extent of the recovery.

-Figure 4b: Similarly to Fig 3b, here I think it would be very interesting to quantify the intensity of the gH2AX signal to know the extent of the recovery.

RESPONSE TO REVIEWERS

GENERAL COMMENTS FOR ALL REVIEWERS

We thank the reviewers for their careful consideration of our manuscript (NCOMMS-22-04401) and for their helpful comments, which have allowed us to substantially improve the study. We found these suggestions to be useful and constructive for the revised manuscript.

To address the points raised by all reviewers, we have performed new experiments and completed new analyses. These additional analyses have solidified the conclusions of the manuscript. Additionally, we have prepared a much-improved text to address the various points raised by reviewers.

Major revisions that address key concerns (many shared by multiple reviewers) are as follows:

1. We performed three independent new experiments and solidified the original conclusion that the X chromosome is transcriptionally reactivated after inhibition of ATR:
 - 1) Gene-specific RNA FISH to detect nascent transcripts of two X-linked genes (*Utx/Kdm6a* and *Lamp2*, new Fig. 1g, h);
 - 2) Cot-1 RNA FISH, which visualizes genome-wide nascent transcription (new supplementary Fig. 3);
 - 3) RNA-sequencing (RNA-seq) using isolated pachytene spermatocytes (new Fig. 2 and new supplementary Fig. 4). Of note, we observed massive transcriptional upregulation of the X and Y chromosomes (82.7- and 64.7- fold increases in mean expression from the X and Y chromosomes, respectively) in isolated pachytene spermatocytes after treatment with ATRi (new Fig. 2b).
2. We further characterized *Setdb1* cKO spermatocytes (recent data have been added to new Supplementary Figs. 9 and 11) and conclude that DDR-mediated initiation of MSCI occurs in *Setdb1* conditional knockout spermatocytes.
3. We added quantification data and statistics for all main figures.
4. We now clarify the nature of the *ex vivo* system we establish here to study MSCI. The foundation for this is the short-term culture method described by the Handel lab (PMID: 19685331)¹. We now clarify confusing sentences and better explain the *ex vivo* system to study MSCI based on the short-term culture method throughout the manuscript.

In addition, we addressed the other points raised by reviewers, as listed below.

Our specific comments to each reviewer are below. Please note: *reviewer comments are in italics*. **Our responses are in bold**. The comments from the reviewers have not been edited. We thank you again for your feedback and consideration.

REVIEWER COMMENTS

Reviewer #1 (Remarks to the Author):

Abe et al. use an ex vivo system to address whether DNA damage response (DDR) is necessary for both initiation and maintenance of meiotic sex chromosome inactivation (MSCI). Using an ATR inhibitor (AZD6738), they find ATR-dependent DDR is necessary for initiation of markers of MSCI (gammaH2AX, PolII exclusion from the XY chromosomes, etc). In addition, after removal of ATR inhibition, these markers are rapidly reestablished, indicating that ATR-dependent regulation of these markers is reversible. They follow up their cell culture experiments with validation in vivo in mice.

This manuscript is a clear, concise exploration of the role of ATR in regulating key factors involved in MSCI. The approaches are logical, the data are very consistent, and the narrative is clear and easy to follow. The follow up with in vivo experiments address concerns that might arise from ex vivo data that conflict with previous studies. The major concern is the authors assumption that gene silencing, and thus MSCI, is also reversed together with the markers of MSCI. This is a huge assumption and there is no data to support that the X is reactivated upon use of the ATR inhibitor. The manuscript would benefit from addressing this issue and modifications to figures, and the experimental approach.

Major:

1. As described above the current writing of the manuscript is that MSCI, and thus gene silencing, is reversible with an ATR inhibitor. This assumption is not supported by any data indicating that X-linked genes are reactivated upon treatment with an ATR inhibitor. The presence of Pol2 or histone PTMs, cytologically, is insufficient in assessing whether there is active gene transcription from the X chromosome. The authors should either rewrite the entire text, eliminating the impact on gene silencing or MSCI, or, preferably, demonstrate that X-linked genes are actively being transcribed after ATRi. To demonstrate X-linked genes are actively transcribed, the authors should detect chromosome-wide nascent transcription (e.g. via GRO-seq) or at a minimum with mRNA-seq combined with RNA-FISH of a few loci.

Thank you for raising this important point to reinforce the main conclusion that the X chromosome is reactivated after the inhibition of ATR. We performed three independent new experiments and solidified this conclusion:

- 1) gene-specific RNA FISH to detect nascent transcripts of two X-linked genes (*Utx/Kdm6a* and *Lamp2*, new Figs. 1g, h),**
- 2) Cot-1 RNA FISH which visualizes genome-wide nascent transcription (new supplementary Fig. 3), and**
- 3) RNA-sequencing (RNA-seq) using isolated pachytene spermatocytes (new Fig. 2, supplementary Fig. 4).**

We decided to analyze *Utx/Kdm6a* and *Lamp2* at 18 Mb and 37 Mb, respectively, on the X chromosome, because these two probes gave very bright signals in our previous experiments^{2,3}, and therefore provide a very sensitive means to evaluate transcriptional activity there. In gene-specific RNA FISH analysis (Fig. 1g and h), *Utx* and *Lamp2* nascent transcripts were detected on the X chromosome in >90% of pachytene spermatocytes only after treatment with ATRi. Additionally, exclusion of Cot-1 signals (nascent transcripts) on the X (normal MSCI) was not observed in >90% of pachytene spermatocytes after treatment with ATRi (new supplementary Fig. 3). We also performed RNA-seq experiments using cultured pachytene spermatocytes with or without treatment

with ATRi and observed massive transcriptional upregulation of the X and Y chromosomes (82.7- and 64.7- fold increases in mean expression from the X and Y chromosomes, respectively) in isolated pachytene spermatocytes after treatment with ATRi (new Fig. 2b).

Together, these data solidify our conclusion that the X chromosome is transcriptionally reactivated after inhibition of ATR.

2. Figure 3b-d: This figure shows loss of reestablishment of MSCl after ATRi is stopped. However, these images alone are indistinguishable from those in which ATRi never occurred. Images from parallel cultures showing establishment first would be more convincing.

We apologize that we did not explain this point clearly in the previous version of the manuscript. Actually, to precisely demonstrate these processes, both now and before, we show pictures from sequential steps from the same experiments in Figs. 1 and Fig. 4. We performed the 24 h culture experiments with ATRi (shown in new Figs. 1c and 1e) and the recovery experiments (new Figs. 4b, d) as sequential experiments. We prepared these samples at the same time using the same testicular cell suspension for Figs. 1 and Fig. 4. Therefore, the data shown in Fig. 1 confirms that DDR signaling was attenuated (MSCl was reverted) by addition of ATRi prior to the reestablishment of MSCl when ATRi was removed (shown in new Fig. 4). We now clarify this point in the Results section (Lines 182-187). Further, in support of new Fig. 4, we include additional representative images (new supplementary Fig. 6a), which show the absence of the γ H2AX domain after 24 h treatment with ATRi prior to the reestablishment of MSCl when ATRi is withdrawn.

3. The authors frequently compare MSCl to X chromosome inactivation, in particular noting that MSCl occurs much more rapidly. This is perhaps not surprising, given that the canonical role of the DDR is to rapidly respond to DNA lesions, whereas X inactivation relies on the expression of an RNA from a single locus, which coats the chromosome starting a cascade of PTM events. The overt comparisons between these two pathways could be reduced to a single comment in the discussion.

We appreciate this suggestion. Accordingly, we have now removed these descriptions from the Introduction and Results sections and added a brief (reduced) comment in the Discussion section (Lines 328-330).

Minor:

1. Figure 1a: The cartoon of the experimental approach should include the drug treatment timeline to clearly explain when ATRi starts and stops in order to analyze mid-late pachytene stage cells.

To clarify this point, we have now added a new schematic to supplementary Fig. 1 that explains the outline of our experiments, and clarifies when ATRi starts and stops; this schematic also explains why we chose to analyze H1T-positive pachytene spermatocytes. As shown in the schematic, H1T-positive pachytene spermatocytes, after 24 hours of culture, have already initiated MSCl at the time of starting culture.

2. All figures: Representative images are provided to show effects of ATRi on localization of BRCA1, exclusion of XY from Poll machinery, etc. As in Figure 1, graphs depicting the quantifications of these data should be presented.

We have now added quantification data for all figures, as follows:

For DDR factors (BRCA1, TOPBP1, ATR, and MDC1): new Fig. 3f
For reestablishment of MSCI (exclusion of Pol II): new Fig. 4e
For histone modifications (SUMO1, Ubiquitination, and H3K27ac): new Fig. 6e
For reversible MSCI in vivo (γ H2AX domain and exclusion of Pol II): new Fig. 8d and f

3. These findings will be of interest to the non-meiotic DDR community, who may not be familiar with MSCI. Thus, a more general introduction into DDR, or their findings in light of non-meiotic DDR pathways in the discussion, would be of greater value to the community. For example, it is not clear what the initial DNA damage signal is that leads to ATR-dependent MSCI, and how factors such as Spo11, ATM, or RPA are involved.

We appreciate this suggestion to clarify the background information for a broad readership, including the non-meiotic DDR community. We revised the Introduction section accordingly (Lines 46-55). Specifically, we now explain how the initial DNA damage signal leads to ATR-dependent MSCI (Lines 52-55) and how factors such as Spo11 and ATM are involved in MSCI (Lines 49-52).

4. The authors references could be more unbiased to their own laboratory's studies. For example, on lines 182+183 the authors reference two studies (Refs. #29 and #30) that found genes "escaping" post-meiotic transcriptional repression in spermatids. Both of these references are from the senior author's laboratory. However, a variety of studies, from multiple groups, have demonstrated sets of genes that "escape" post-meiotic repression. Or for example, "Silencing of unsynapsed meiotic chromosomes in the mouse" (Turner et al., 2005) is a landmark publication with direct relevance to these studies, establishing the lack of synapse as a mechanism for meiotic silencing, however it is not referenced once in the manuscript.

Thank you for catching this point. We now include citations for three additional references that demonstrate sets of genes that "escape" post-meiotic repression (Mueller et al., 2008 Nat Genet, Mueller et al., 2013 Nat Genet, Moretti et al., 2016 Epigenetics Chromatin)^{4, 5, 6} (Line 229).

We also agree that Turner et al., 2005⁷ is the landmark publication that establishes the lack of synapse as a mechanism for meiotic silencing. We now cite this paper, along with another independent study that establishes the lack of synapse as a mechanism for meiotic silencing (Baarends et al., 2005 Mol Cell Biol)⁸ and Dr. John Schimenti's commentary that coined the concept of meiotic silencing of unsynapsed chromatin (MSUC: Schimenti 2005 Nat Genet)⁹ (Line 47). Additionally, we now introduce two other previous studies that show the involvement of the DDR pathway in MSCI (Turner et al., 2004 Curr Biol, Femandes-Capetillo et al., 2003 Dev Cell)^{10, 11} (Line 49).

Reviewer #2 (Remarks to the Author):

In mammals, homologs that fail to synapse during meiosis are transcriptionally inactivated. This process of transcriptional inactivation or meiotic silencing, drives inactivation of the heterologous XY bivalent in male germ cells and is named meiotic sex chromosome inactivation [MSCI]. MSCI is hypothesized to work as a meiotic surveillance mechanism in meiosis. During the last decade, mouse genetic analysis has established that ATR (within the DDR) directs MSCI (Royo et al., 2013; PMID: 23824539). It has been established that ATR first regulates HORMAD1/2 phosphorylation and localization of BRCA1 and TOPBP1 at unsynapsed axes. In more advanced states, ATR transduce the signal through MDC1 and H2AFX. And finally, ATR catalyzes histone H2AFX phosphorylation (γ H2AX formation) and this promotes through TRIM28 the SETDB1-dependent deposition of H3K9me3, XY condensation, and gene silencing (Hirota et al., 2018; PMID: 30393076).

In this MS, Abe et al., make use of a pharmacological inhibition of ATR to study the known functions of ATR in this pathway during a short time window and in a reversible manner using a short-term spermatocyte culture protocol. They conclude that two layers of modification are operating in the MSCI pathway one reversible (γ H2AX) and one irreversible (H3K9me3). They conclude that active ATR is the primary mechanism of silencing in MSCI.

There are several criticisms to the MS by Abe et al.

-The authors state in the text and in the abstract that they have established a novel ex vivo system to study MSCI and demonstrate that active DDR signaling is required to initiate and maintain MSCI.

The first point is that the short-term ex vivo cell culture they have used is not a novel system as it has been applied in lot of previous occasions since it was described by the Handel lab (PMID: 19685331) as a modification of the previous method (PMID: 3651542). If the method is indeed new, they should compare its performance in comparison with the already published.

We thank this reviewer for constructive criticism to improve the manuscript. For the first point, we apologize if our explanation was misleading. One of our main points is that we enabled the examination/manipulation of MSCI by taking advantage of, and adapting, the culture method described by the Handel lab (PMID: 19685331)¹, thereby establishing a system to study MSCI *ex vivo*. In particular, our method uses a single-cell suspension of all spermatogenic cells from testes, as is now clarified in the Methods section (Lines 410-414). This is a major technical difference from the method developed by the Handel lab, which uses isolated pachytene spermatocytes only¹. The previous system is based upon labor-intensive STA-PUT-based isolation of pachytene spermatocytes. It would take ~ 6 hours for cell isolation prior to cell culture. However, our system introduces cells into culture faster (immediately after preparation of single-cell suspensions and therefore takes ~ 1 hour) because it cuts several cell isolation steps.

In this revision, we clearly acknowledge the culture method described by the Handel lab (PMID: 19685331)¹ as the technical foundation of our study. Further, we now clarify potentially confusing sentences and explain throughout the manuscript the novelty of our system to study MSCI *ex vivo* (in culture) in the Abstract Line 27, Introduction Line 73, Results Line 89-91, Discussion Line 315; for example, in the Abstract we added the following phrase “Here we establish a system to study MSCI *ex vivo*, based on a short-term culture method” (Line 27). Additionally, in this revision, we now provide a battery of solid evidence that MSCI can be manipulable in our *ex vivo* system (as validated by gene-

specific RNA FISH, Cot-1 RNA FISH, and RNA-seq in new Figs. 1 and 2). Together, these results validate the capacity of our system to study MSCI *ex vivo* in culture.

We do want to point out that, in this revision, to perform RNA-seq using a homogenous population, we newly performed the culture of FACS-isolated pachytene spermatocytes (similar to the paper from the Handel lab) with or without treatment with ATRi (new Fig.2). This condition yields a similar result to our above-mentioned method using a single-cell suspension of all cells from testes, based on the reversible formation of the γ H2AX domain on the XY (new supplementary Fig.4). These independent analyses confirm that MSCI can be manipulated in culture. We now clarify this point in the Results section (Lines 130-136).

We also appreciate the reviewer pointing out the old method (O'Brian, Biol Reprod 1987, PMID: 3651542)¹². We have now added this citation along with a previous study of the Handel lab (Handel et al., Dev Genet 1995)¹³ when we introduce the method (Lines 90-91).

Two major issues must be considered in assessing the function of spermatocytes under these ex vivo conditions; i) that they lack their surrounding Sertoli cells and are no longer interconnected with other spermatocytes in large syncytia and that as consequence ii) the culture / experimental setting cannot exceed 24h. This is not trivial and constitutes a serious limitation to the whole study. There is an alternative cell culture condition that allow longer culturing of seminiferous tubules in air-liquid interface. This alternative is not plausible since it would require experimental design of the whole work.

In the original protocol written by the Handel lab, they mentioned that the culture can be done for more than 24 h. They stated that “despite viability limited to 24–36 h, the methods described here have proven useful to examine and manipulate key steps in late meiotic prophase”. Indeed, our culture method is not limited to a 24 h period, and here we show that 24 h incubation and an additional 3 h incubation (27 h total culture) is feasible; cells were viable after 24 h incubation and were able to reestablish MSCI during the 3h window. In addition, the short-term culture method has been extensively used in many other studies in the field, including some milestone papers^{14, 15}. Altogether, our research strategy was validated with our data and supported by ample literature.

Nevertheless, we acknowledge the possible limitation of the culture method in that spermatocytes do not have a direct association with other cells such as Sertoli cells and are not interconnected with other spermatocytes in large syncytia. We recognize this point in the discussion (Line 384-387).

The other point is that it was already known that ATR is needed to promote MSCI. In this regard, the authors always refer to the analysis of the DDR pathway throughout the paper when they indeed only modify / alter / study the activity of ATR through its pharmacological inhibition making use mainly of the short-term culture of spermatocytes. The only genetic approach carried out by the authors in the MS (cKO of SETDB1) has already been analyzed previously making use of a similar transgenic CRE (Hirota et al., 2018; PMID: 30393076).

We agree that genetic mouse models are powerful tools to elucidate molecular functions. However, particularly in the meiosis field, the genetic approaches also have a serious technical limitation: mutation of critical meiotic factors, such as DDR factors, leads to defective MSCI and complete germ cell elimination at the onset of the mid-pachytene stage by the pachytene checkpoint. Therefore, it is still unclear how those factors

contribute to the maintenance of MSCI after it is established at the onset of the pachytene stage. Our study fills this knowledge gap primarily using a cell culture model. The advantage of the culture method is that it enables manipulation of critical factors in wild-type spermatocytes after normal MSCI is established. By taking advantage of this, we were able to distinguish the initiation and maintenance mechanisms, and we uncovered a critical function for ATR in the maintenance of MSCI. Thus, the utilization of a cell culture model has enabled the conclusion that MSCI must be actively maintained by the ATR-DDR pathway. This point is significant because germ cell death takes place in *Atr*-cKO spermatocytes, precluding the analysis of ATR's function in the maintenance of MSCI after its establishment using mouse models.

The authors conclude that the presence of γ H2AX in the absence of H3K9Me3 leads to meiotic silencing. However, the results shown in figure 4C are not demonstrative (there is no a clear exclusion of Pol II immunolabeling) and in order to raise such a conclusion several additional more robust molecular procedures should have been used to firmly proof this ascertain.

Regarding the analysis of previous Figure 4 (now Figure 5C), we now added quantification of signal intensity and demonstrated that γ H2AX signals and Pol II intensity were mutually exclusive, showing that Pol II signals are excluded from regions of γ H2AX signals.

With regard to the *Setdb1*-cKO phenotype, we have now carefully evaluated whether MSCI takes place in *Setdb1*-cKO spermatocytes (in the absence of H3K9me3). Because chromosome synapsis defects were observed in *Setdb1*-cKO pachytene spermatocytes (~ 40% in *Setdb1*-cKO early pachytene spermatocytes in Hirota et al.¹⁶, and ~50 % in *Setdb1*-cKO early- to mid-pachytene spermatocytes in Cheng et al.¹⁷), we evaluated MSCI in pachytene spermatocytes that completed chromosome synapsis. This feature was detected by the presence of a single linear HORMAD1 domain (HORMAD1 only shows a single linear domain detecting unsynapsed XY axes after the completion of normal autosomal synapsis). This is a valid analysis because autosomal asynapsis disrupts MSCI¹⁸, and we can exclude the possibility that MSCI is indirectly disrupted due to autosomal asynapsis. In these *Setdb1*-cKO pachytene spermatocytes, which have complete synapsis, we find that MSCI takes place; Cot-1 RNA FISH signals were excluded from the XY (in >90% of *Setdb1*-cKO cells (new supplementary Figs. 11b, c), and gene-specific RNA FISH revealed that two X-linked genes, *Utx* and *Lamp2*, are not actively transcribed (in >90% of *Setdb1*-cKO cells (new supplementary Figs. 11d, e). We decided to analyze *Utx* (at 18Mb on the X) and *Lamp2* (at 37Mb on the X) because these two probes gave very bright signals in our previous experiments^{2,3}, and therefore provide a very sensitive means to evaluate transcriptional activity there. We validated the use of these probes to examine MSCI (new Figs. 1g and h) and conclude that DDR-mediated initiation of MSCI occurs in *Setdb1* cKO spermatocytes.

By contrast, the RNAseq results by the Turner lab show active transcription of inactive genes in the spermatocytes of the cSETDB1KO encoded by X and Y chromosomes lacking H3K9me3 ((between 3.3 and 3.7 times, respectively) and also by direct in situ hybridization of sex-linked RNA probes in spermatocytes (PMID: 20360682, Figure 5) despite showing γ H2AX labelling in the XY axes and in their corresponding chromatin (sex body) (see Figure 3a in PMID: 30393076).

In our new RNA-seq analysis, disruption of MSCI (by treatment with ATRi) leads to massive transcriptional upregulation of the X and Y chromosomes (82.7- and 64.7- fold

increases in mean expression from the X and Y chromosomes, respectively) in isolated pachytene spermatocytes (new Fig. 2). By contrast, Hirota et al.¹⁶ reported that *Setdb1*-cKO pachytene spermatocytes exhibited 3.3- and 3.7- fold upregulation for the X and Y chromosomes. Although we cannot directly compare these studies due to various technical differences, our finding of differences of more than an order of magnitude supports the notion that the DDR is the direct (central) mechanism for MSCI initiation and maintenance. Still, our results suggest that H3K9me3 could possibly have a role in bolstering DDR-dependent MSCI. It should be noted that in our cytological experiments, we only evaluated MSCI in the *Setdb1*-cKO pachytene spermatocytes that completed autosomal synapsis. One possibility could be that MSCI is indirectly disrupted in *Setdb1*-cKO pachytene spermatocytes that have autosomal asynapsis. We now mention these possibilities in the Discussion section (Lines 368-372).

In an unpublished study, we have identified MCAF2 (ATF7IP2) as the meiosis-specific regulator of SETDB1 and H3K9me3. In *Mcaf2* knockout pachytene spermatocytes, SETDB1 and H3K9me3 were mislocalized from the sex chromosomes, and X and Y-linked genes were mildly upregulated akin to the *Setdb1*-cKO phenotype. These data support the possibility that SETDB1-dependent H3K9me3 may bolster DDR-dependent MSCI. We plan to further clarify these mechanisms in our future studies.

Taking as a whole, the study is mostly descriptive and is mostly based on chemical inhibition of ATR (with the obvious limitations) and lack functional and mechanistic analysis. This is a strong limitation together with the lack of novelty or reduced impact of most of the descriptions carried out in the MS.

In this revision, we extensively performed new functional and mechanistic analyses and presented a battery of evidence that solidifies the original conclusions. The novelty of this study is that, by taking advantage of a system we develop here to study MSCI *ex vivo*, we filled a major knowledge gap that could not have been addressed using mouse genetic models. Importantly, our *ex vivo* model yielded key mechanistic insights into MSCI by revealing it as a dynamic and reversible process. Another notable mechanistic insight is that DDR-dependent MSCI takes place in the absence of SETDB-dependent H3K9me3.

Other comments:

The number of animals and cells analyzed should be specified for each experiment of the different figures. The results should always be quantified and statistically analyzed.

The number of animals and cells analyzed are now described in each figure panel or legend. Also, we added quantification data for all figures as follows:

For DDR factors (BRCA1, TOPBP1, ATR, and MDC1): new Fig. 3f

For reestablishment of MSCI (exclusion of Pol II): new Fig. 4e

For histone modifications (SUMO1, Ubiquitination, and H3K27ac): new Fig. 6e

For reversible MSCI *in vivo* (γ H2AX domain and exclusion of Pol II): new Fig. 8d and f

Reviewer #3 (Remarks to the Author):

In this manuscript of the Namekawa lab, the authors study the role of the DDR in the establishment and maintenance of the meiotic sex chromosome inactivation (MSCI). To do so, they developed an ex vivo system that allows them to analyze MSCI and prove that active DDR signaling is required to start and maintain MSCI in a dynamic and reversible process. In this work, the authors describe reversible-and irreversible-PTM modifications linked to MSCI and demonstrate that the DDR is responsible for silencing the sex chromosomes during meiosis independently of H3K9me3.

This work represents a significant advance in understanding how sex chromosomes are silenced during the meiotic prophase. Furthermore, it describes a novel ex vivo system that permits the dynamic study of mammalian meiotic prophase in a way that has not been done before and might be of great utility to overcome the intrinsic limitations of the study of mammalian meiosis. The logic behind each experiment is well presented. The quality of the data displayed is very high, and the presented data seem to support the conclusions raised by the authors. Nonetheless, I have several major and minor comments I would like the authors to address before publications:

Major comments:

*-The main comment I have about this study is the use of immunostaining against the RNA polymerase II as the single readout of gene expression. I understand that this marker has been extensively used to indicate sex chromosome inactivation, but I do not think the absence of RNA pol II signal on the sex chromosomes is sufficient to conclude that MSCI works appropriately. In my opinion, the authors should also perform RNA-FISH against particular XY genes (e.g., *Scml2*, *Zfx*, ...) to show that they are adequately silenced.*

Thank you for your valuable comments. To better evaluate MSCI, following suggestions by this and other reviewers, we performed three independent new experiments and thereby solidified the conclusion that the X is reactivated after treatment with ATRi:

- 1) Gene-specific RNA FISH to detect nascent transcripts of two X-linked genes (*Utx/Kdm6a* and *Lamp2*: new Figs. 1g, h);**
- 2) Cot-1 RNA FISH, which visualizes genome-wide nascent transcription (new supplementary Fig. 3);**
- 3) RNA-sequencing (RNA-seq) using isolated pachytene spermatocytes (new Fig. 2 and new supplementary Fig. 4).**

We decided to analyze *Utx/Kdm6a* and *Lamp2* at 18 Mb and 37 Mb on the X chromosome, respectively, because these two probes gave very bright signals in our previous experiments^{2,3}, and therefore provide a very sensitive means to evaluate transcriptional activity there. In gene-specific RNA FISH analysis (Figs. 1g and h), *Utx* and *Lamp2* nascent transcripts were detected on the X chromosome in >90% of pachytene spermatocytes only after treatment with ATRi. Further, exclusion of Cot-1 signals (nascent transcripts) on the X (normal MSCI) was not observed in >90% of pachytene spermatocytes after treatment with ATRi (new supplementary Fig. 3). In addition, we performed RNA-seq experiments using cultured pachytene spermatocytes with or without treatment with ATRi and demonstrate that X and Y-linked genes were reactivated chromosome-wide: 247 genes among 339 expressed X-linked genes were upregulated, and 5 genes among 9 expressed Y-linked genes were upregulated after ATRi-treatment (new Fig. 2 and new supplementary Fig. 4). Together, these data solidify our conclusion that the X chromosome is transcriptionally reactivated after the inhibition of ATR.

-I think the authors should make an effort to analyze the outcome of the experiments they perform more objectively. Only a minority of the figures (e.g., 1b, d, f) have some quantification of the data observed and statistical analysis of those. Although the images provided in most of the figures are clear, objective quantification of the results should be provided for each of the experiments performed (e.g., Figures 2, 3d, 4, 5, 7, S1b, S2, S3, S4, and S5).

As suggested, we have added quantification data for all main figures as follows:

For DDR factors (BRCA1, TOPBP1, ATR, and MDC1): new Fig. 3f

For reestablishment of MSCI (exclusion of Pol II): new Fig. 4e

For histone modifications (SUMO1, Ubiquitination, and H3K27ac): new Fig. 6e

For reversible MSCI in vivo (γ H2AX domain and exclusion of Pol II): new Fig. 8d and f

Because we were able to present quantification data for all main figures and thereby confirm the main aspects of our conclusions, we have decided to present representative pictures for supplementary figures.

Minor comments:

-Line 75: Please explain a little bit more the novelty of this ex vivo system and how it relates to previous culture systems used to study ATR function in meiosis (Pacheco et al. 2018; Sims et al. 2021)

The major novelty of our ex vivo system is that we were able to evaluate gene silencing in MSCI in culture, specifically during the maintenance phase of MSCI. This is a critical knowledge gap that could not have been filled with previous mouse models, although these mouse models uncovered the molecular mechanism of the initial steps of MSCI.

Previous ex vivo work using ATRi focuses on other aspects in meiotic prophase I. Pacheco et al. combined organ culture and AZ20 inhibitor treatment and revealed critical roles for ATR in homologous recombination. Sims et al. performed proteomics screening and identified potential phosphorylation targets by ATR. By contrast, our study reports how MSCI is maintained. We have now clarified these differences in the Discussion section (Line 319-324).

-I would suggest to the authors to include the information displayed in supplemental figure 4 in Figure 5

We appreciate this suggestion. However, because of the many other additions to the main figures in this revision, we decided to keep these data in new supplementary Fig. 8.

-Line 216: I think authors should provide PAS-H stainings to show better the meiotic arrest occurring in Setdb1-cKO and, if possible, perform a TUNEL assay on these sections.

Thank you for this suggestion. We added hematoxylin & eosin (H&E) staining data (instead of PAS-H staining) and a TUNEL assay with quantification as new supplementary Fig. 9.

-Line 217: I do not think the IF against Setdb1 is enough to show the reduction of Setdb1 expression on conditional mutants. I want to propose that the authors perform a western blot analysis of the whole testis protein extracts to certify the status of the conditional mutant.

In this revision, we added further analysis of the *Setdb1* cKO mouse model to confirm the conditional deletion. We used germ cell-specific *Ddx4*-Cre to induce conditional deletion of *Setdb1*, as described in Cheng et al.¹⁷. Our *Ddx4*-Cre *Setdb1* cKO mouse model showed a consistent phenotype as reported in Cheng et al.¹⁷ (new supplementary Fig. 9), although we were not able to perform Western blots as suggested due to the limited number of mutant testes available.

-Figure 2: The shape of the XY chromosomes varies during meiotic prophase and can be used to stage spermatocytes (Page et al., 2012). As the pachytene stage progresses, the X chromosomes curls. Thus, early pachynema cells have long, straight X chromosomes, and mid/late pachynema cells have mostly curly X chromosomes. The X and Y chromosomes are displayed differently in the cartoons provided for control and AZ20-treated samples. While in the controls, the X is curled; in the treated samples, the X is straight. I wonder if this is drawn on purpose and if it reflects a different conformation of the XY chromosomes in these two populations.

Thank you for catching this error. We did not intend to explain the confirmational difference. We found that the XY configuration was not patently changed. We clarified the model (now in new Fig. 3b-e).

-Figure 3b: In this case, I think it is very relevant to quantify the intensity of the γ H2AX signal to know the extent of the recovery.

Thank you for this suggestion. We have now quantified γ H2AX intensity, and show the ratios of γ H2AX intensity on an XY chromosomal area and an autosomal area for each nucleus (new supplementary Fig. 6b). γ H2AX on the XY was decreased after treatment with ATRi (10 μ M) to a basal level (from 16.0 ± 2.1 to 2.4 ± 0.19 : mean \pm s.e.m) and recovered after additional incubation for 3 h after removal of ATRi (recovered to 14.0 ± 1.3 : mean \pm s.e.m). These data are largely consistent with the data shown in new Figure 4 (previous Fig. 3B).

-Figure 4b: Similarly to Fig 3b, here I think it would be very interesting to quantify the intensity of the γ H2AX signal to know the extent of the recovery.

Thank you for pointing this out. We agree that it would be useful to trace reactivation of the DDR after removing the ATR inhibitor. Because the pattern of γ H2AX on the XY axis after ATRi-removal is varied, as shown in new Fig. 5b, c, we decided to show signal quantification along arbitrary lines (tracks) across individual nuclei (new Fig. 5c).

References for this section:

1. La Salle S, Sun F, Handel MA. Isolation and short-term culture of mouse spermatocytes for analysis of meiosis. *Methods Mol Biol* **558**, 279-297 (2009).
2. Namekawa SH, Payer B, Huynh KD, Jaenisch R, Lee JT. Two-step imprinted X inactivation: repeat versus genic silencing in the mouse. *Mol Cell Biol* **30**, 3187-3205 (2010).
3. Che L, Alavattam KG, Stambrook PJ, Namekawa SH, Du C. BRUCE preserves genomic stability in the male germline of mice. *Cell Death Differ* **27**, 2402-2416 (2020).
4. Mueller JL, Mahadevaiah SK, Park PJ, Warburton PE, Page DC, Turner JM. The mouse X chromosome is enriched for multicopy testis genes showing postmeiotic expression. *Nat Genet* **40**, 794-799 (2008).
5. Mueller JL, *et al.* Independent specialization of the human and mouse X chromosomes for the male germ line. *Nat Genet* **45**, 1083-1087 (2013).
6. Moretti C, Vaiman D, Tores F, Cocquet J. Expression and epigenomic landscape of the sex chromosomes in mouse post-meiotic male germ cells. *Epigenetics Chromatin* **9**, 47 (2016).
7. Turner JM, *et al.* Silencing of unsynapsed meiotic chromosomes in the mouse. *Nat Genet* **37**, 41-47 (2005).
8. Baarends WM, *et al.* Silencing of unpaired chromatin and histone H2A ubiquitination in mammalian meiosis. *Mol Cell Biol* **25**, 1041-1053 (2005).
9. Schimenti J. Synapsis or silence. *Nat Genet* **37**, 11-13 (2005).
10. Turner JM, *et al.* BRCA1, histone H2AX phosphorylation, and male meiotic sex chromosome inactivation. *Curr Biol* **14**, 2135-2142 (2004).
11. Fernandez-Capetillo O, *et al.* H2AX is required for chromatin remodeling and inactivation of sex chromosomes in male mouse meiosis. *Dev Cell* **4**, 497-508 (2003).
12. O'Brien DA. Stage-specific protein synthesis by isolated spermatogenic cells throughout meiosis and early spermiogenesis in the mouse. *Biol Reprod* **37**, 147-157 (1987).
13. Handel MA, Caldwell KA, Wiltshire T. Culture of pachytene spermatocytes for analysis of meiosis. *Dev Genet* **16**, 128-139 (1995).
14. Rao HB, *et al.* A SUMO-ubiquitin relay recruits proteasomes to chromosome axes to regulate meiotic recombination. *Science* **355**, 403-407 (2017).
15. Guan Y, *et al.* SKP1 drives the prophase I to metaphase I transition during male meiosis. *Sci Adv* **6**, eaaz2129 (2020).
16. Hirota T, *et al.* SETDB1 Links the Meiotic DNA Damage Response to Sex Chromosome Silencing in Mice. *Dev Cell* **47**, 645-659.e646 (2018).

17. Cheng EC, *et al.* The Essential Function of SETDB1 in Homologous Chromosome Pairing and Synapsis during Meiosis. *Cell Rep* **34**, 108575 (2021).
18. Mahadevaiah SK, Bourc'his D, de Rooij DG, Bestor TH, Turner JM, Burgoyne PS. Extensive meiotic asynapsis in mice antagonises meiotic silencing of unsynapsed chromatin and consequently disrupts meiotic sex chromosome inactivation. *J Cell Biol* **182**, 263-276 (2008).

We again thank the reviewers for their insightful comments, which helped us significantly improve our study. We appreciate your consideration and hope that our publication is now suitable for publication in Nature Communications.

**Most sincerely,
Satoshi Namekawa and Hironori Abe**

REVIEWERS' COMMENTS

Reviewer #1 (Remarks to the Author):

1. In Figure 1h (and elsewhere) they quantitate "percent without FISH signal". It would be simpler to present the inverse data: Percent with FISH signal.

2. Figure 4b-d: While the authors have explained that the establishment of MSCI is presented earlier in Figure 1, and that the experiments were sequential, it is still preferable to show establishment, loss, and reestablishment in a single figure, as that is the complete experiment. Figures should stand alone; the reader shouldn't have to read the text to understand that the first part of the experiment is in an earlier figure.

Reviewer #2 (Remarks to the Author):

The revised version of the MS has been improved substantially with the new experiments that the authors have carried out following the reviewer's reports. All my concerns have been addressed successfully with explanations, modifications in the text and new convincing experiments.

Reviewer #3 (Remarks to the Author):

The authors have satisfactorily addressed most of my comments and modified the manuscript improving it significantly. Thus, I recommend this manuscript for publication.

RESPONSE TO REVIEWERS

We thank the reviewers for their careful consideration of our manuscript (NCOMMS-22-04401A). We also thank Reviewer #1 for their additional suggestions. We found these suggestions to be useful the final version of the manuscript.

Our specific comments to each reviewer are below. Please note: *reviewer comments are in italics*. **Our responses are in bold**. The comments from the reviewers have not been edited. We thank you again for your feedback and consideration.

Reviewer #1 (Remarks to the Author):

1. In Figure 1h (and elsewhere) they quantitate "percent without FISH signal". It would be simpler to present the inverse data: Percent with FISH signal.

We agree with your suggestion. We inverted and replaced the nascent RNA data (Fig. 1h, Supplementary Fig. 11e).

2. Figure 4b-d: While the authors have explained that the establishment of MSC1 is presented earlier in Figure 1, and that the experiments were sequential, it is still preferable to show establishment, loss, and reestablishment in a single figure, as that is the complete experiment. Figures should stand alone; the reader shouldn't have to read the text to understand that the first part of the experiment is in an earlier figure.

We agree with your suggestion. We moved the previous Supplementary Fig. 6a (defective DDR after 24 h incubation with ATRi) to main Fig. 4b. Now, Fig. 4 presents all critical sequential data.